# Effector target-guided engineering of an integrated domain expands the disease resistance profile of a rice NLR immune receptor

Josephine HR Maidment[1†], Motoki Shimizu[2], Adam R Bentham[1], Sham Vera[1‡], Marina Franceschetti[1], Apinya Longya[1§], Clare EM Stevenson[1], Juan Carlos De la Concepcion[1#], Aleksandra Białas[3], Sophien Kamoun[3], Ryohei Terauchi[2,4], Mark J Banfield[1*]

[1]Department of Biological Chemistry, John Innes Centre, Norwich, United Kingdom; [2]Division of Genomics and Breeding, Iwate Biotechnology Research Center, Iwate, Japan; [3]The Sainsbury Laboratory, University of East Anglia, Norwich Research Park, Norwich, United Kingdom; [4]Laboratory of Crop Evolution, Graduate School of Agriculture, Kyoto University, Kyoto, Japan

*For correspondence:
Mark.banfield@jic.ac.uk

Present address: [†]PHIM Plant Health Institute, University of Montpellier, INRAE, CIRAD, Institut Agro, IRD, Montpellier, France; [‡]ETH Zurich, Institute of Molecular Plant Biology, Plant Evolutionary Genetics Group, Zürich, Switzerland; [§]Department of Genetics, Faculty of Science, Kasetsart University, Bangkok, Thailand; [#]Gregor Mendel Institute of Molecular Plant Biology, Austrian Academy of Sciences, Vienna, Austria

**Abstract** A subset of plant intracellular NLR immune receptors detect effector proteins, secreted by phytopathogens to promote infection, through unconventional integrated domains which resemble the effector's host targets. Direct binding of effectors to these integrated domains activates plant defenses. The rice NLR receptor Pik-1 binds the *Magnaporthe oryzae* effector AVR-Pik through an integrated heavy metal-associated (HMA) domain. However, the stealthy alleles AVR-PikC and AVR-PikF avoid interaction with Pik-HMA and evade host defenses. Here, we exploited knowledge of the biochemical interactions between AVR-Pik and its host target, OsHIPP19, to engineer novel Pik-1 variants that respond to AVR-PikC/F. First, we exchanged the HMA domain of Pikp-1 for OsHIPP19-HMA, demonstrating that effector targets can be incorporated into NLR receptors to provide novel recognition profiles. Second, we used the structure of OsHIPP19-HMA to guide the mutagenesis of Pikp-HMA to expand its recognition profile. We demonstrate that the extended recognition profiles of engineered Pikp-1 variants correlate with effector binding in planta and in vitro, and with the gain of new contacts across the effector/HMA interface. Crucially, transgenic rice producing the engineered Pikp-1 variants was resistant to blast fungus isolates carrying AVR-PikC or AVR-PikF. These results demonstrate that effector target-guided engineering of NLR receptors can provide new-to-nature disease resistance in crops.

## Editor's evaluation

Engineering NLR proteins to improve disease resistance in crop plants is a major goal of the field. This study applies knowledge from structural and evolutionary studies of the rice NLR protein Pik-1 and cognate effector protein AVR-Pik to engineering of new disease resistance genes. The authors nicely demonstrate that it is indeed possible to engineer resistance proteins with broad recognition specificity for the rice blast fungus. The work is of interest to colleagues in synthetic biology, protein engineering and plant-pathogen interactions.

## Introduction

Intracellular nucleotide-binding and leucine-rich repeat (NLR) domain-containing immune receptors are essential components of the plant innate immune system (*Jones and Dangl, 2006*; *Jones et al., 2016*). These receptors detect effector proteins which are delivered into host cells by invading pathogens and pests to promote virulence. NLR receptors have a modular domain architecture typically consisting of an N-terminal coiled-coil (CC or $CC_R$) or Toll/Interleukin-1 receptor (TIR) domain, a central nucleotide-binding (NB-ARC) domain, and a C-terminal leucine-rich repeat (LRR) domain. In addition, some NLRs contain non-canonical domains which are integrated into the protein architecture either at the N- or C-termini or between canonical domains (*Sarris et al., 2016*; *Cesari et al., 2014*; *Kroj et al., 2016*; *Bailey et al., 2018*). These integrated domains (IDs) resemble effector virulence targets and either directly bind or are modified by effector proteins to activate NLR-mediated immune signaling (*Cesari et al., 2014*; *Oikawa et al., 2020*; *Maidment et al., 2021*; *Maqbool et al., 2015*; *Sarris et al., 2015*; *Mukhi et al., 2021*).

Many of the characterized resistance (R) genes used to confer disease resistance in crop breeding programs encode NLR proteins (*Kourelis and van der Hoorn, 2018*). However, NLR-mediated resistance can be overcome through silencing or deletion of effectors in pathogen genomes, the gain of new effectors or effector functions, or mutation to evade NLR activation (*Raffaele et al., 2010*; *Yoshida et al., 2016*). Engineering NLRs to detect currently unrecognized effector proteins would provide new opportunities to control plant pathogens. Early attempts to engineer NLRs focused on random mutagenesis followed by gain-of-function screening, with some success in both expanding recognition profiles to new effector variants and increasing the sensitivity of the receptor (*Giannakopoulou et al., 2015*; *Segretin et al., 2014*; *Farnham and Baulcombe, 2006*; *Harris et al., 2013*). More recently, modification of the effector target PBS1, which is guarded by the NLR protein RPS5 led to successful engineering of novel recognition by this system (*Kim et al., 2016*; *Pottinger et al., 2020*; *Pottinger and Innes, 2020*; *Helm et al., 2019*). RPS5 is activated by cleavage of the *Arabidopsis thaliana* protein kinase PBS1 by the *Pseudomonas syringae* effector AvrPphB. By varying the PBS1 cleavage site, the RPS5/PBS1 system has been engineered to recognize proteases from different pathogens (*Pottinger et al., 2020*; *Helm et al., 2019*). Using a different strategy, a protein domain targeted for degradation by the phytoplasma effector SAP05 was fused to the C-terminus of the TIR-NLR RRS1-R. RRS1-R represses the immune cell death-triggering activity of a second TIR-NLR, RPS4. While transient co-expression of the engineered RRS1-R, RPS4, and the phytoplasma effector SAP05 led to cell death in *N. tabacum*, transgenic *A. thaliana* plants were not resistant to phytoplasma carrying the SAP05 effector (*Wang et al., 2021*).

Integrated domains can facilitate the recognition of structure- and sequence-diverse effectors which target similar host proteins. This is exemplified by the TIR-NLR pair RRS1 and RPS4, which mediate recognition of the structurally distinct effectors PopP2 from *Ralstonia solanacearum* and AvrRps4 from *Pseudomonas syringae* pv. *pisi* (*Sarris et al., 2015*; *Mukhi et al., 2021*; *Deslandes et al., 2003*; *Saucet et al., 2015*; *Huh et al., 2017*; *Le Roux et al., 2015*). Furthermore, the potential to replace naturally occurring integrated domains with nanobodies of defined specificity to confer disease resistance has recently been demonstrated (*Kourelis et al., 2023*). Modification of existing integrated domains, or the incorporation of entirely new protein domains into an NLR structure could deliver new recognition specificities and extend the toolbox of resistance genes available to combat crop pathogens and pests.

The paired rice CC-NLR proteins Pik-1 and Pik-2 cooperatively activate plant defense in response to the blast pathogen effector AVR-Pik (*Ashikawa et al., 2008*; *Kanzaki et al., 2012*; *Zdrzałek et al., 2020*). The sensor NLR Pik-1 contains an integrated HMA domain between the CC and NB-ARC domains (*Maqbool et al., 2015*, *Figure 1—figure supplement 1a*). Direct binding of AVR-Pik to the HMA domain is required to activate Pik-mediated immunity (*Maqbool et al., 2015*; *De la Concepcion et al., 2018*; *De la Concepcion et al., 2021*). Multiple Pik alleles have been described in different rice cultivars, with most amino acid polymorphisms located within the integrated HMA domain of Pik-1. Five Pik alleles (Pikp, Pikm, Pikh, Piks, and Pik*) have been functionally characterized for their response to blast isolates carrying different AVR-Pik variants (*Maqbool et al., 2015*; *Kanzaki et al., 2012*; *De la Concepcion et al., 2018*; *De la Concepcion et al., 2021*, *Figure 1—figure supplement 1c*). To date, six AVR-Pik variants (A-F) have been described, which differ in five amino acid positions at the HMA-binding interface (*Kanzaki et al., 2012*; *Yoshida et al., 2009*; *Longya et al., 2019*, *Figure 1—figure*

*supplement 1b*). These polymorphisms influence the binding of the effector to the integrated HMA domain of Pik-1 (*Maqbool et al., 2015*; *De la Concepcion et al., 2018*; *Longya et al., 2019*). Interestingly, the Asp67 and Lys78 polymorphisms of AVR-PikC and AVR-PikF, respectively, disrupt interactions between the effector and all tested integrated Pik-HMA domains (*Maidment et al., 2021*; *De la Concepcion et al., 2021*; *Longya et al., 2019*). To date, none of the characterized Pik alleles can confer disease resistance to blast isolates carrying AVR-PikC or AVR-PikF (*Maqbool et al., 2015*; *Kanzaki et al., 2012*; *De la Concepcion et al., 2018*; *De la Concepcion et al., 2021*).

The molecular basis of interaction between AVR-Pik effectors and the integrated HMA domains of Pikp-1, Pikm-1, and Pikh-1 has been well explored (*Maqbool et al., 2015*; *De la Concepcion et al., 2018*; *De la Concepcion et al., 2021*). Pikp-1 is only able to recognize the AVR-PikD variant, however, the introduction of two amino acid changes (Asn261Lys and Lys262Glu) extends recognition to AVR-PikE and AVR-PikA, phenocopying the recognition profile of Pikm-1 and Pikh-1 (*De la Concepcion et al., 2019*).

The NLR pair RGA5 and RGA4 detect the blast pathogen effectors AVR-Pia and AVR1-CO39, with activation requiring binding of the effector to an integrated HMA domain at the C-terminus of RGA5 (*Okuyama et al., 2011*; *Cesari et al., 2013*; *Ortiz et al., 2017*). Crystal structures of the RGA5-HMA/AVR1-CO39 and Pik-HMA/AVR-Pik complexes were used to engineer the RGA5-HMA domain to bind AVR-PikD in addition to its cognate effectors AVR-Pia and AVR1-CO39 and deliver cell death in *Nicotiana benthamiana*, but not disease resistance in transgenic rice (*Cesari et al., 2022*). More recently, RGA5 has been engineered to bind the non-cognate effector AVR-Pib. Transgenic rice carrying the engineered RGA5 variant was resistant to AVR-Pib-expressing *M. oryzae* strains, with resistance comparable to that displayed by the (untransformed) rice cultivar K14 which carries the Pib CC-NLR resistance gene (*Liu et al., 2021*). These studies demonstrate the potential for engineering integrated domains to alter the recognition profile of the NLR protein, however, engineering new-to-nature effector recognition is yet to be reported.

The AVR-Pik effector targets members of the rice heavy metal-associated isoprenylated plant protein (HIPP) and heavy metal-associated plant protein (HPP) families through direct interaction with their HMA domain, supporting the hypothesis that NLR integrated domains are likely derived from host proteins (*Oikawa et al., 2020*; *Maidment et al., 2021*). In a previous study, we showed that all AVR-Pik effector variants bind to the HMA domain of OsHIPP19 with high affinity and elucidated the structural basis of this interaction by determining the crystal structure of a OsHIPP19-HMA/AVR-PikF complex (*Maidment et al., 2021*). This shows that effector variants that are not bound by Pik-HMA domains, and escape immune recognition, retain a tight binding for HMA domains of their putative host targets.

Here, we leverage our understanding of the interaction between OsHIPP19 and AVR-Pik to engineer the integrated HMA domain of Pik-1 to expand recognition to the stealthy AVR-PikC and AVR-PikF variants, enabling new-to-nature disease resistance profiles in an NLR. We use two parallel strategies to engineer recognition. First, we demonstrate that exchanging the HMA domain of Pikp-1 for that of OsHIPP19 (including additional amino acid substitutions to prevent autoactivity), gives a chimeric Pik-1 which binds AVR-Pik effectors and triggers AVR-PikC- and AVR-PikF-dependent cell death in *N. benthamiana*. Second, guided by the structure of the OsHIPP19-HMA/AVR-PikF complex, we use targeted mutagenesis of Pikp-1 to give a second engineered Pik-1 receptor capable of binding to AVR-PikC and AVR-PikF and triggering cell death in *N. benthamiana*. Finally, we show that transgenic rice expressing either of these engineered Pik-1 proteins is resistant to blast pathogen strains carrying AVR-PikC or AVR-PikF, while rice expressing wild-type Pikp is susceptible. This work highlights how a biochemical and structural understanding of the interaction between a pathogen effector and its host target can guide the rational engineering of NLR proteins with the novel, and new-to-nature, disease resistance profiles.

## Results

### A Pikp-1<sup>OsHIPP19</sup> chimera extends binding and response to previously unrecognized AVR-Pik variants

Previously, we reported that all AVR-Pik variants, including AVR-PikC and AVR-PikF, bind to the HMA domain of OsHIPP19 with high affinity (*Maidment et al., 2021*). The HMA domains of Pikp-1 and

OsHIPP19 share 51% amino acid identity and are structurally similar; the RMSD (as calculated in Coot using secondary structure matching) between Pikp-HMA (PDB 6G10) and OsHIPP19-HMA (PDB 7B1I) is 0.97 Å across 71 amino acids. We hypothesized that exchanging the HMA domain of Pikp-1 for the HMA domain of OsHIPP19 would result in an NLR capable of binding and responding to AVR-PikC and AVR-PikF.

For this exchange, amino acids 188–263 (inclusive) of Pikp-1 were replaced with amino acids 2–77 of OsHIPP19 to give the chimeric NLR protein Pikp-1[OsHIPP19] (*Figure 1—figure supplement 2a*). To test whether Pikp-1[OsHIPP19] could associate with AVR-Pik effector variants in planta, we performed co-immunoprecipitation experiments in *N. benthamiana*. Each of the myc-AVR-Pik variants (an N-terminal myc tag was used for effectors in all experiments in *N. benthamiana*) was transiently co-expressed with Pikp-1[OsHIPP19]-HF (C-terminal 6xHis/3xFLAG tag, used for all Pikp-1 constructs expressed in *N. benthamiana*) by agroinfiltration. Pikp-2 was not included to prevent the onset of cell death, which reduces protein levels in the plant cell extract, and previous work has shown that AVR-Pik can associate with Pik-1 in the absence of Pik-2 (*De la Concepcion et al., 2018*). Following immunoprecipitation with anti-FLAG beads to enrich for Pikp-1[OsHIPP19], western blot analysis showed that all AVR-Pik variants, except AVR-PikF, co-precipitated with Pikp-1[OsHIPP19] (*Figure 1—figure supplement 3*). As a control, and consistent with previous studies, AVR-PikD associated with Pikp-1, while AVR-PikC did not.

We then transiently co-expressed epitope-tagged Pik-1, Pikp-2, and AVR-Pik in *N. benthamiana* using cell death as a proxy for immune activation (*Maqbool et al., 2015*; *De la Concepcion et al., 2018*; *De la Concepcion et al., 2021*; *Longya et al., 2019*). We found that when co-expressed with Pikp-2-HA (C-terminal hemagglutinin tag, used for all Pikp-2 constructs expressed in *N. benthamiana*), Pikp-1[OsHIPP19] is autoactive and triggers spontaneous cell death in the absence of the effector (*Figure 1a and b*, *Figure 1—figure supplement 4a*). This autoactivity requires an intact P-loop and MHD motif in Pikp-2, as cell death is abolished when Pikp-1[OsHIPP19] is transiently co-expressed with either Pikp-2[K217R] or Pikp-2[D559V] (*Figure 1—figure supplement 5a*, *Figure 1—figure supplement 5b*, *Figure 1—figure supplement 6*). Cell death was reduced, but not abolished, when Pikp-1[OsHIPP19] with a Lys296Arg mutation in the P-loop motif (Pikp-1[OsHIPP19_K296R]) was transiently co-expressed with Pikp-2 (*Figure 1—figure supplement 5a*, *Figure 1—figure supplement 5b*, *Figure 1—figure supplement 6*). Western blot analysis indicated that all fusion proteins were produced (*Figure 1—figure supplement 5c*, *Figure 1—figure supplement 7a*).

A previous study showed that autoactivity following HMA domain exchange could be abolished by reverting the degenerate metal-binding motif of the HMA domain ('MxCxxC') to the corresponding amino acids in Pikp-1 (*Białas et al., 2021*). Based on this observation, we exchanged seven amino acids (encompassing the entire MxCxxC motif) in the β1-α1 loop of the Pikp-1[OsHIPP19] chimera for the corresponding amino acids in Pikp-1 (*Figure 1—figure supplement 2b*, *Figure 1—figure supplement 8*). The resulting chimera, Pikp-1[OsHIPP19-mbl7] hereafter (mbl7 refers to seven amino acids in the 'metal-binding loop'), was not autoactive, and did not trigger spontaneous cell death in the absence of the effector. Crucially, Pikp-1[OsHIPP19-mbl7] retained the ability to trigger cell death in *N. benthamiana* when co-expressed with Pikp-2 and AVR-PikD. Further, Pikp-1[OsHIPP19-mbl7] also triggered cell death in *N. benthamiana* when co-expressed with Pikp-2 and AVR-PikC or AVR-PikF (*Figure 1c–f*, *Figure 1—figure supplement 4b–c*, *Figure 1—figure supplement 9*). Western blot analysis showed that all proteins for these cell death assays were produced in leaf tissue (*Figure 1—figure supplement 7b*, *Figure 1—figure supplement 7c*). We also showed that Pikp-1[OsHIPP19-mbl7] triggered cell death in *N. benthamiana* when co-expressed with Pikp-2 and AVR-PikA, AVR-PikE (*Figure 1—figure supplement 10*, *Figure 1—figure supplement 11*). Finally, we confirmed that Pikp-1[OsHIPP19-mbl7] retains binding to AVR-Pik variants by co-immunoprecipitation in *N. benthamiana*, including AVR-PikF (*Figure 1g*).

## Structure-guided mutagenesis of Pikp-1 extends response to previously unrecognized AVR-Pik variants

Alongside the HMA-domain exchange strategy, we also used a structure-guided approach to target point mutations in Pikp-HMA that could extend the effector recognition profile of Pikp without triggering autoimmunity.

The interaction surfaces between integrated Pik-HMA domains and AVR-Pik effectors are well-characterized, with crystal structures revealing three predominant interfaces (termed 1–3) between the proteins (*Maqbool et al., 2015*; *De la Concepcion et al., 2018*; *De la Concepcion et al., 2021*;

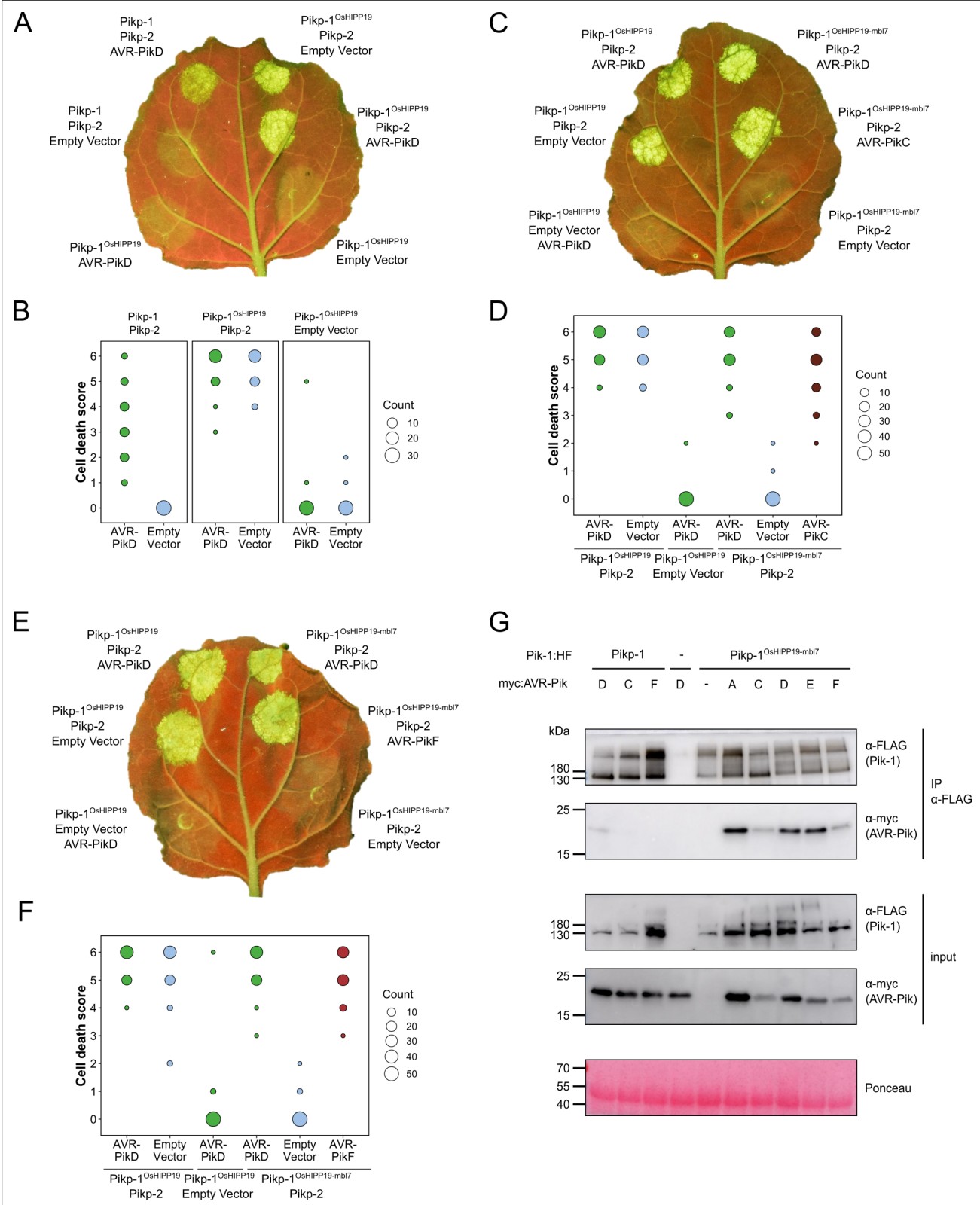

**Figure 1.** The Pikp-1$^{OsHIPP19-mbl7}$ chimera expands binding and response to previously unrecognized AVR-Pik effector variants. (**A**) Representative leaf image showing the Pikp-1$^{OsHIPP19}$ chimera is autoactive in *N. benthamiana* in a Pikp-2 dependent manner. Nucleotide-binding and leucine-rich repeat (NLR)-mediated responses appear as autofluorescence imaged under UV light. Pikp-mediated response to AVR-PikD (positive control, top left), Pikp-1$^{OsHIPP19}$/Pikp-2 without effector shows autoactivity (top right), Pikp-1$^{OsHIPP19}$/Pikp-2 response remains in the presence of AVR-PikD (middle right).

*Figure 1 continued on next page*

*Figure 1 continued*

Other leaf positions represent relevant negative controls. (**B**) Pikp-mediated response scoring represented as dot plots to summarize 30 repeats of the experiment shown in (**A**) across three independent experiments (Materials and methods, *Figure 1—figure supplement 4a*). Fluorescence intensity is scored as previously described (*Maqbool et al., 2015*; *De la Concepcion et al., 2018*). (**C**) The Pikp-1$^{OsHIPP19-mbl7}$ chimera does not display autoactive cell death in *N. benthamiana* (bottom right), as seen for Pikp-1$^{OsHIPP19}$ (middle left), but retains response to AVR-PikD and expands Pikp-mediated response to AVR-PikC (middle right). (**D**) Pikp-mediated response scoring represented as dot plots to summarize 60 repeats of the experiment shown in (**C**) across three independent experiments (Materials and methods, *Figure 1—figure supplement 4b*). (**E**) As for (**C**), but showing the expanded Pikp-mediated response to AVR-PikF. (**F**) As for (**D**) but for 60 repeats of the experiment in (**E**) across three independent experiments (Materials and methods, *Figure 1—figure supplement 4c*). (**G**) Western blots following co-immunoprecipitation reveal that the Pikp-1$^{OsHIPP19-mbl7}$ chimera associates with all AVR-Pik effector variants in *N. benthamiana*. Plant cell lysates were probed for the expression of Pikp-1/Pikp-1$^{OsHIPP19-mbl7}$ and AVR-Pik effector variants using anti-FLAG and anti-Myc antiserum, respectively. Total protein extracts were visualized by Ponceau Staining.

The online version of this article includes the following source data and figure supplement(s) for figure 1:

**Source data 1.** Unedited and uncropped blot for panel G, IP Pik-1 α-FLAG, Pik-1 α-FLAG, with relevant bands labeled.

**Source data 2.** Unedited and uncropped blot for panel G, IP Pik-1 α-FLAG, Pik-1 α-FLAG.

**Source data 3.** Unedited and uncropped blot for panel G, IP Pik-1 α-FLAG, AVRPik α-myc, with relevant bands labeled.

**Source data 4.** Unedited and uncropped blot for panel G, IP Pik-1 α-FLAG, AVRPik α-myc.

**Source data 5.** Unedited and uncropped blot for panel G, input, Pik-1 α-FLAG, with relevant bands labeled.

**Source data 6.** Unedited and uncropped blot for panel G, input, Pik-1 α-FLAG.

**Source data 7.** Unedited and uncropped blot for panel G, input, AVRPik α-myc, with relevant bands labeled.

**Source data 8.** Unedited and uncropped blot for panel G, input, AVRPik α-myc.

**Source data 9.** Unedited and uncropped blot for panel G, Ponceau stain, with relevant bands labeled.

**Source data 10.** Unedited and uncropped blot for panel G, Ponceau stain.

**Source data 11.** Cell death scores used for dot plots in panel B.

**Source data 12.** Cell death scores used for dot plots in panel D.

**Source data 13.** Cell death scores used for dot plots in panel F.

**Figure supplement 1.** Domain structure and resistance profiles for the proteins in this study.

**Figure supplement 2.** Schematic representation of the Pikp-1$^{OsHIPP19}$ (**A**) and Pikp-1$^{OsHIPP19-mbl7}$ (**B**) chimeras.

**Figure supplement 3.** Western blots following co-immunoprecipitation show that the Pikp-1$^{OsHIPP19}$ chimera binds to all AVR-Pik effector variants except, surprisingly, AVR-PikF in *N. benthamiana*.

**Figure supplement 3—source data 1.** Unedited and uncropped blot, IP Pik-1 α-FLAG, Pik-1 α-FLAG, with relevant bands labelled.

**Figure supplement 3—source data 2.** Unedited and uncropped blot, IP Pik-1 α-FLAG, Pik-1 α-FLAG.

**Figure supplement 3—source data 3.** Unedited and uncropped blot, IP Pik-1 α-FLAG, AVRPik α-myc, with relevant bands labeled.

**Figure supplement 3—source data 4.** Unedited and uncropped blot, IP Pik-1 α-FLAG, AVRPik α-myc.

**Figure supplement 3—source data 5.** Unedited and uncropped blot, input, Pik-1 α-FLAG, with relevant bands labeled.

**Figure supplement 3—source data 6.** Unedited and uncropped blot, input, Pik-1 α-FLAG.

**Figure supplement 3—source data 7.** Unedited and uncropped blot, input, AVRPik α-myc, with relevant bands labeled.

**Figure supplement 3—source data 8.** Unedited and uncropped blot, input, AVRPik α-myc.

**Figure supplement 3—source data 9.** Unedited and uncropped blot, Ponceau stain, with relevant bands labeled.

**Figure supplement 3—source data 10.** Unedited and uncropped blot, Ponceau stain.

**Figure supplement 4.** Pikp-mediated response scoring is represented as dot plots, subdivided by replicate, for repeats of experiments presented in *Figure 1a, c and e* (panels A, B, and C; respectively).

**Figure supplement 5.** The Pikp-1$^{OsHIPP19}$ chimera requires both the P-loop and MHD motifs in Pikp-2 for autoactivity, and the P-loop in Pikp-1 for full cell death.

**Figure supplement 5—source data 1.** Unedited and uncropped blot for panel C, Pik-1 α-FLAG, with relevant bands labeled.

**Figure supplement 5—source data 2.** Unedited and uncropped blot for panel C, Pik-1 α-FLAG.

**Figure supplement 5—source data 3.** Unedited and uncropped blot for panel C, Pik-2 α-HA, with relevant bands labeled.

**Figure supplement 5—source data 4.** Unedited and uncropped blot for panel C, Pik-2 α-HA.

**Figure supplement 5—source data 5.** Unedited and uncropped blot for panel C, Ponceau stain, with relevant bands labeled.

**Figure supplement 5—source data 6.** Unedited and uncropped blot for panel C, Ponceau stain.

**Figure supplement 5—source data 7.** Cell death scores used for dot plots for panel B.

*Figure 1 continued on next page*

*Figure 1 continued*

**Figure supplement 6.** Pikp-mediated response scoring is represented as dot plots, subdivided by replicate, for repeats of the experiment presented in *Figure 1—figure supplement 5a*.

**Figure supplement 7.** Western blots confirm the accumulation of proteins in *N. benthamiana* for the cell death assays shown in *Figure 1*.

**Figure supplement 7—source data 1.** Unedited and uncropped blot for panel A, Pik-1 α-FLAG, with relevant bands labeled.

**Figure supplement 7—source data 2.** Unedited and uncropped blot for panel A, Pik-1 α-FLAG.

**Figure supplement 7—source data 3.** Unedited and uncropped blot for panel A, Pik-2 α-HA, with relevant bands labeled.

**Figure supplement 7—source data 4.** Unedited and uncropped blot for panel A, Pik-2 α-HA.

**Figure supplement 7—source data 5.** Unedited and uncropped blot for panel A, AVR-Pik α-myc, with relevant bands labeled.

**Figure supplement 7—source data 6.** Unedited and uncropped blot for panel A, AVR-Pik α-myc.

**Figure supplement 7—source data 7.** Unedited and uncropped blot for panel A, Ponceau stain, with relevant bands labeled.

**Figure supplement 7—source data 8.** Unedited and uncropped blot for panel A, Ponceau stain.

**Figure supplement 7—source data 9.** Unedited and uncropped blot for panel B, Pik-1 α-FLAG, with relevant bands labeled.

**Figure supplement 7—source data 10.** Unedited and uncropped blot for panel B, Pik-1 α-FLAG.

**Figure supplement 7—source data 11.** Unedited and uncropped blot for panel B, Pik-2 α-HA, with relevant bands labeled.

**Figure supplement 7—source data 12.** Unedited and uncropped blot for panel B, Pik-2 α-HA.

**Figure supplement 7—source data 13.** Unedited and uncropped blot for panel B, AVR-Pik α-myc, with relevant bands labeled.

**Figure supplement 7—source data 14.** Unedited and uncropped blot for panel B, AVR-Pik α-myc.

**Figure supplement 7—source data 15.** Unedited and uncropped blot for panel B, Ponceau stain, with relevant bands labeled.

**Figure supplement 7—source data 16.** Unedited and uncropped blot for panel B, Ponceau stain.

**Figure supplement 7—source data 17.** Unedited and uncropped blot for panel C, Pik-1 α-FLAG, with relevant bands labeled.

**Figure supplement 7—source data 18.** Unedited and uncropped blot for panel C, Pik-1 α-FLAG.

**Figure supplement 7—source data 19.** Unedited and uncropped blot for panel C, Pik-2 α-HA, with relevant bands labeled.

**Figure supplement 7—source data 20.** Unedited and uncropped blot for panel C, Pik-2 α-HA.

**Figure supplement 7—source data 21.** Unedited and uncropped blot for panel C, AVR-Pik α-myc, with relevant bands labeled.

**Figure supplement 7—source data 22.** Unedited and uncropped blot for panel C, AVR-Pik α-myc.

**Figure supplement 7—source data 23.** Unedited and uncropped blot for panel C.

**Figure supplement 7—source data 24.** Unedited and uncropped blot for panel C, Ponceau stain.

**Figure supplement 8.** Location of the β1-α1 loop (brown) in Pikp-HMA (blue) is distant from the effector (green) binding surface in the crystal structure of complexes between these proteins.

**Figure supplement 9.** Statistical analysis by estimation methods of the cell death assays presented in *Figure 1*, for Pikp-1$^{OsHIPP19-mbl7}$/Pikp-2 with (**A**) AVR-PikD, AVR-PikC and empty vector, and (**B**) AVR-PikD, AVR-PikF, and empty vector.

**Figure supplement 10.** The Pikp-1$^{OsHIPP19-mbl7}$ chimera responds to AVR-PikE and AVR-PikA.

**Figure supplement 10—source data 1.** Cell death scores used for dot plots in panel B.

**Figure supplement 11.** Pikp-mediated response scoring is represented as dot plots, subdivided by replicate, for repeats of the experiment presented in *Figure 1—figure supplement 10a*.

*De la Concepcion et al., 2019*). These interfaces are also observed in the structure of the HMA domain of OsHIPP19 in complex with AVR-PikF (PDB accession code 7B1I *Maidment et al., 2021*). The Asp67 and Lys78 polymorphisms that distinguish AVR-PikC and AVR-PikF from AVR-PikE and AVR-PikA, respectively, are located at interface 2. In the crystal structure of Pikh-HMA/AVR-PikC, the side chain of AVR-PikC$^{Asp67}$ extends towards a loop in the HMA domain containing Pikh-HMA$^{Asp224}$. This loop is shifted away from the effector, likely due to steric clash and/or repulsion by the two Asp sidechains, and intermolecular hydrogen bonds between Pikh-HMA$^{Asp224}$ and AVR-PikC$^{Arg64}$ are disrupted. We hypothesized that compensatory mutations at interface 2 could mitigate against the disruption caused by AVR-PikC$^{Asp67}$. Therefore, we introduced Asp224Ala and Asp224Lys mutations in the Pikp-1$^{NK-KE}$ background (*De la Concepcion et al., 2019*) and tested these constructs in cell death assays in *N. benthamiana*. Neither mutation extended the Pikp-1$^{NK-KE}$-mediated cell death response to AVR-PikC, and both mutations reduced the extent of the response to AVR-PikD (*Figure 2—figure supplement 1*, *Figure 2—figure supplement 2*).

Pikh-1 and Pikp-1[NK-KE] differ from Pikp-1 by one and two amino acids, respectively, at interface 3. These amino acid differences are sufficient to extend binding and cell death response to AVR-PikE and AVR-PikA, even though the residues that distinguish these variants from AVR-PikD are located at interface 2. We, therefore, predicted that we could engineer a modified Pik-1 that interacts with AVR-PikC/AVR-PikF by mutating other interfaces in the HMA domain to compensate for disruption at the site of the polymorphic residue. The crystal structure of the OsHIPP19-HMA/AVR-PikF complex revealed additional hydrogen bond interactions at interface 3 relative to integrated HMAs in the complex with AVR-Pik variants (*Maidment et al., 2021*). The side chain of OsHIPP19[Glu72] was particularly striking. The corresponding residue in all described Pik-1 HMA domains is serine, and while the hydroxyl group of the serine side chain only forms an intramolecular hydrogen bond within the HMA domain, the bulkier OsHIPP19[Glu72] side chain extends across the interface and forms a direct hydrogen bond with the effector (*Figure 2a*). We, therefore, introduced a Ser258Glu mutation in the Pikp-1[NK-KE] background to give the triple mutant Pikp-1[SNK-EKE] and tested the ability of this protein to respond to AVR-Pik variants in *N. benthamiana* cell death assays. Firstly, we confirmed that Pikp-1[SNK-EKE] was not autoactive, evidenced by a lack of cell death following co-expression with Pikp-2 only (in the absence of effectors). Next, we established that Pikp-1[SNK-EKE] remained functional and caused cell death on co-expression with Pikp-2 and AVR-PikD. Crucially, transient expression of Pikp-1[SNK-EKE], but not Pikp-1[NK-KE], with Pikp-2 and either AVR-PikC or AVR-PikF, triggered cell death suggestive of new effector response specificities (*Figure 2b–e*, *Figure 2—figure supplement 3*, *Figure 2—figure supplement 4*). Western blot analysis indicated that all fusion proteins were produced (*Figure 2—figure supplement 5*). Pikp-1[NK-KE] has previously been shown to cause cell death when co-expressed with Pikp-2 and AVR-PikE or AVR-PikA, and consistent with these findings, Pikp-1[SNK-EKE] also triggered cell death when co-expressed with Pikp-2 and either of these effector variants (*Figure 2—figure supplement 6*, *Figure 2—figure supplement 7*).

We tested whether the Ser258Glu mutation alone was sufficient to extend the cell death response to AVR-PikC or AVR-PikF using the cell death assay. When Pikp-1[S258E] was co-infiltrated with Pikp-2 and either AVR-PikC or AVR-PikF no cell death was observed (*Figure 2—figure supplement 8*, *Figure 2—figure supplement 9*), demonstrating that the triple mutation is necessary for response to these effectors.

## The Ser258Glu mutation extends the binding of Pikp-HMA[NK-KE] to AVR-PikC and AVR-PikF in vitro

The extent of the Pik/AVR-Pik-dependent cell death response in *N. benthamiana* largely correlates with binding affinity in vitro and in planta (*Maqbool et al., 2015*; *De la Concepcion et al., 2018*; *De la Concepcion et al., 2021*; *De la Concepcion et al., 2019*). To test whether the Pikp-1[SNK-EKE] response to AVR-PikC or AVR-PikF in *N. benthamiana* correlates with increased binding to the modified HMA domain, we first used surface plasmon resonance (SPR) with purified proteins. Pik-HMA domains and AVR-Pik effectors were purified from *E. coli* cultures using established protocols for the production of these proteins (*Maidment et al., 2021*; *Maqbool et al., 2015*; *De la Concepcion et al., 2018*; *De la Concepcion et al., 2021*; *De la Concepcion et al., 2019*). For SPR, AVR-Pik effector variants were immobilized on a $Ni^{2+}$-NTA sensor chip via a C-terminal 6xHis tag. Pikp-HMA, Pikp[NK-KE]-HMA or Pikp[SNK-EKE]-HMA has flowed over the surface of the chip at three different concentrations (4 nM, 40 nM, and 100 nM). The binding ($R_{obs}$, measured in response units, RU) was recorded and expressed as a percentage of the maximum theoretical responses (%Rmax), assuming a 2:1 HMA:effector interaction model (*Maqbool et al., 2015*). Consistent with previous studies, Pikp-HMA did not bind AVR-PikC (nor AVR-PikF), and weak binding was observed for Pikp[NK-KE]-HMA to AVR-PikC (and also to AVR-PikF). By contrast, Pikp[SNK-EKE]-HMA bound to both AVR-PikC and AVR-PikF with higher apparent affinity (larger %Rmax) than Pikp[NK-KE]-HMA (*Figure 3a–b*). This result was consistent across the three concentrations investigated, though binding (and %Rmax) was low for all three HMA domains at 4 nM (*Figure 3—figure supplement 1*).

## The Ser258Glu mutation extends the binding of Pikp-HMA[NK-KE] to AVR-PikC and AVR-PikF in planta

Next, we determined whether the Ser258Glu mutation also extends binding to AVR-PikC and AVR-PikF in the full-length NLR in planta using co-immunoprecipitation. Full-length Pikp-1, Pikp-1[NK-KE], and

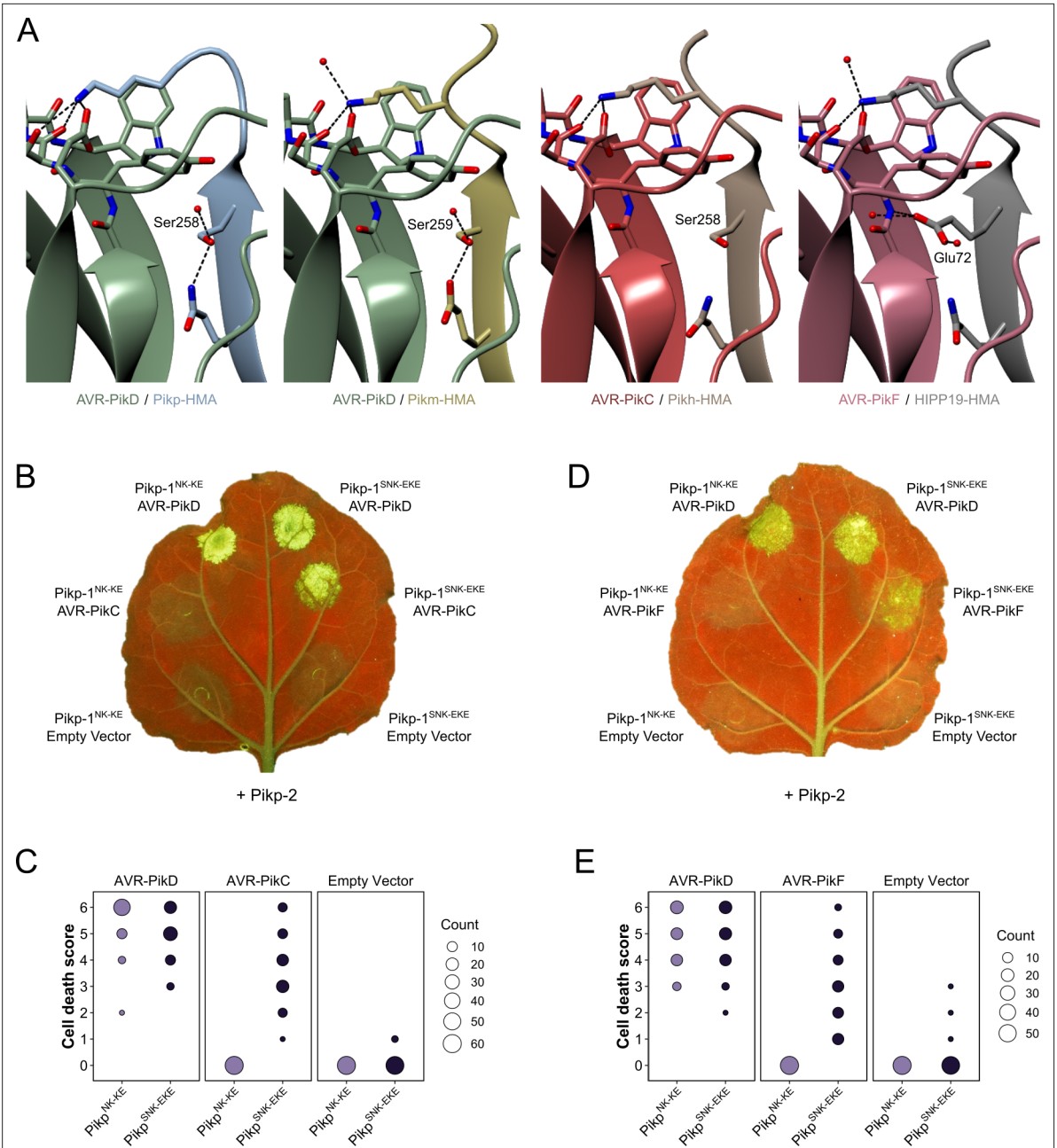

**Figure 2.** Structure-guided mutagenesis of Pikp-1 expands response to previously unrecognized AVR-Pik effector variants. (**A**) Comparison of the crystal structures of AVR-Pik effector variants in complex with Pik-HMA domains (PDB entries 6G10, 6FU9, and 7A8X) and AVR-PikF in complex OsHIPP19 (PDB entry 7B1I) suggests the addition of an S258E mutation to the NK-KE mutations described previously (*De la Concepcion et al., 2019*) could introduce new contacts across the protein:protein interface. Protein structures are represented as ribbons with relevant side chains displayed as cylinders. Dashed lines indicate hydrogen bonds. Relevant water molecules are represented as red spheres. (**B**) The Pikp$^{SNK-EKE}$ mutant gains response to AVR-PikC (right, middle) where no response is observed for Pikp$^{NK-KE}$ (left, middle). Further, the Pikp$^{SNK-EKE}$ mutant is not autoactive (right, bottom) and retains response to AVR-PikD (right, top). All infiltration spots contain Pikp-2. (**C**) Pikp-mediated response scoring represented as dot plots to summarize 60 repeats of the experiment shown in (**B**) across three independent experiments (Materials and methods, *Figure 2—figure supplement 3a*). (**D**) and (**E**) as described for (**B**) and (**C**) but with AVR-PikF and 57 repeats across three independent experiments (Materials and methods, *Figure 2—figure supplement 3b*).

The online version of this article includes the following source data and figure supplement(s) for figure 2:

**Source data 1.** Cell death scores used for dot plots in panel C.

**Source data 2.** Cell death scores used for dot plots in panel E.

**Figure supplement 1.** The mutations D224A and D224K mutations in the Pikp$^{NK-KE}$ background do not extend response to AVR-PikC.

*Figure 2 continued on next page*

*Figure 2 continued*

**Figure supplement 1—source data 1.** Unedited and uncropped blot for panel C, Pik-1 α-FLAG, with relevant bands labeled.

**Figure supplement 1—source data 2.** Unedited and uncropped blot for panel C, Pik-1 α-FLAG.

**Figure supplement 1—source data 3.** Unedited and uncropped blot for panel C, Pik-2 α-HA, with relevant bands labeled.

**Figure supplement 1—source data 4.** Unedited and uncropped blot for panel C, Pik-2 α-HA.

**Figure supplement 1—source data 5.** Unedited and uncropped blot for panel C, AVR-Pik α-myc, with relevant bands labeled.

**Figure supplement 1—source data 6.** Unedited and uncropped blot for panel C, AVR-Pik α-myc.

**Figure supplement 1—source data 7.** Unedited and uncropped blot for panel C, Ponceau stain, with relevant bands labeled.

**Figure supplement 1—source data 8.** Unedited and uncropped blot for panel C, Ponceau stain.

**Figure supplement 1—source data 9.** Cell death scores used for dot plots in panel B.

**Figure supplement 2.** Pikp-mediated response scoring is represented as dot plots, subdivided by replicate, for repeats of the experiment presented in *Figure 2—figure supplement 1a*.

**Figure supplement 3.** Pikp-mediated response scoring represented as dot plots, subdivided by replicate, for repeats of the experiment presented in *Figure 2b* (**A**) and 2d (**B**).

**Figure supplement 4.** Statistical analysis by estimation methods of the cell death assays presented in *Figure 2*, for Pikp-1$^{NK-KE}$/Pikp-2 and Pikp-1$^{SNK-EKE}$/Pikp-2 with (**A**) AVR-PikC and (**B**) AVR-PikF.

**Figure supplement 5.** Western blots confirm the accumulation of proteins in *N. benthamiana* for the cell death assays shown in *Figure 2*.

**Figure supplement 5—source data 1.** Unedited and uncropped blot for panel A, Pik-1 α-FLAG, with relevant bands labeled.

**Figure supplement 5—source data 2.** Unedited and uncropped blot for panel A, Pik-1 α-FLAG.

**Figure supplement 5—source data 3.** Unedited and uncropped blot for panel A, Pik-2 α-HA, with relevant bands labeled.

**Figure supplement 5—source data 4.** Unedited and uncropped blot for panel A, Pik-2 α-HA.

**Figure supplement 5—source data 5.** Unedited and uncropped blot for panel A, AVR-Pik α-myc, with relevant bands labeled.

**Figure supplement 5—source data 6.** Unedited and uncropped blot for panel A, AVR-Pik α-myc.

**Figure supplement 5—source data 7.** Unedited and uncropped blot for panel A.

**Figure supplement 5—source data 8.** Unedited and uncropped blot for panel A, Ponceau stain.

**Figure supplement 5—source data 9.** Unedited and uncropped blot for panel B, Pik-1 α-FLAG, with relevant bands labeled.

**Figure supplement 5—source data 10.** Unedited and uncropped blot for panel B, Pik-1 α-FLAG.

**Figure supplement 5—source data 11.** Unedited and uncropped blot for panel B, Pik-2 α-HA, with relevant bands labeled.

**Figure supplement 5—source data 12.** Unedited and uncropped blot for panel B, Pik-2 α-HA.

**Figure supplement 5—source data 13.** Unedited and uncropped blot for panel B, AVR-Pik α-myc, with relevant bands labeled.

**Figure supplement 5—source data 14.** Unedited and uncropped blot for panel B, AVR-Pik α-myc.

**Figure supplement 5—source data 15.** Unedited and uncropped blot for panel B, Ponceau stain, with relevant bands labeled.

**Figure supplement 5—source data 16.** Unedited and uncropped blot for panel B, Ponceau stain.

**Figure supplement 6.** Pikp-1$^{SNK-EKE}$ chimera responds to AVR-PikE and AVR-PikA.

**Figure supplement 6—source data 1.** Cell death scores used for dot plots in panel B.

**Figure supplement 7.** Pikp-mediated response scoring is represented as dot plots, subdivided by replicate, for repeats of the experiment presented in *Figure 2—figure supplement 6a*.

**Figure supplement 8.** The Pikp S258E mutation alone does not extend response to AVR-PikC or AVR-PikF.

**Figure supplement 8—source data 1.** Unedited and uncropped blot for panel C, Pik-1 α-FLAG, with relevant bands labeled.

**Figure supplement 8—source data 2.** Unedited and uncropped blot for panel C, Pik-1 α-FLAG.

**Figure supplement 8—source data 3.** Unedited and uncropped blot for panel C, Pik-2 α-HA, with relevant bands labeled.

**Figure supplement 8—source data 4.** Unedited and uncropped blot for panel C, Pik-2 α-HA.

**Figure supplement 8—source data 5.** Unedited and uncropped blot for panel C, AVR-Pik α-myc, with relevant bands labeled.

**Figure supplement 8—source data 6.** Unedited and uncropped blot for panel C, AVR-Pik α-myc.

**Figure supplement 8—source data 7.** Unedited and uncropped blot for panel C, Ponceau stain, with relevant bands labeled.

**Figure supplement 8—source data 8.** Unedited and uncropped blot for panel C, Ponceau stain.

**Figure supplement 8—source data 9.** Cell death scores used for dot plots in panel B.

**Figure supplement 9.** Pikp-mediated response scoring is represented as dot plots, subdivided by replicate, for repeats of the experiment presented in *Figure 2—figure supplement 8a*.

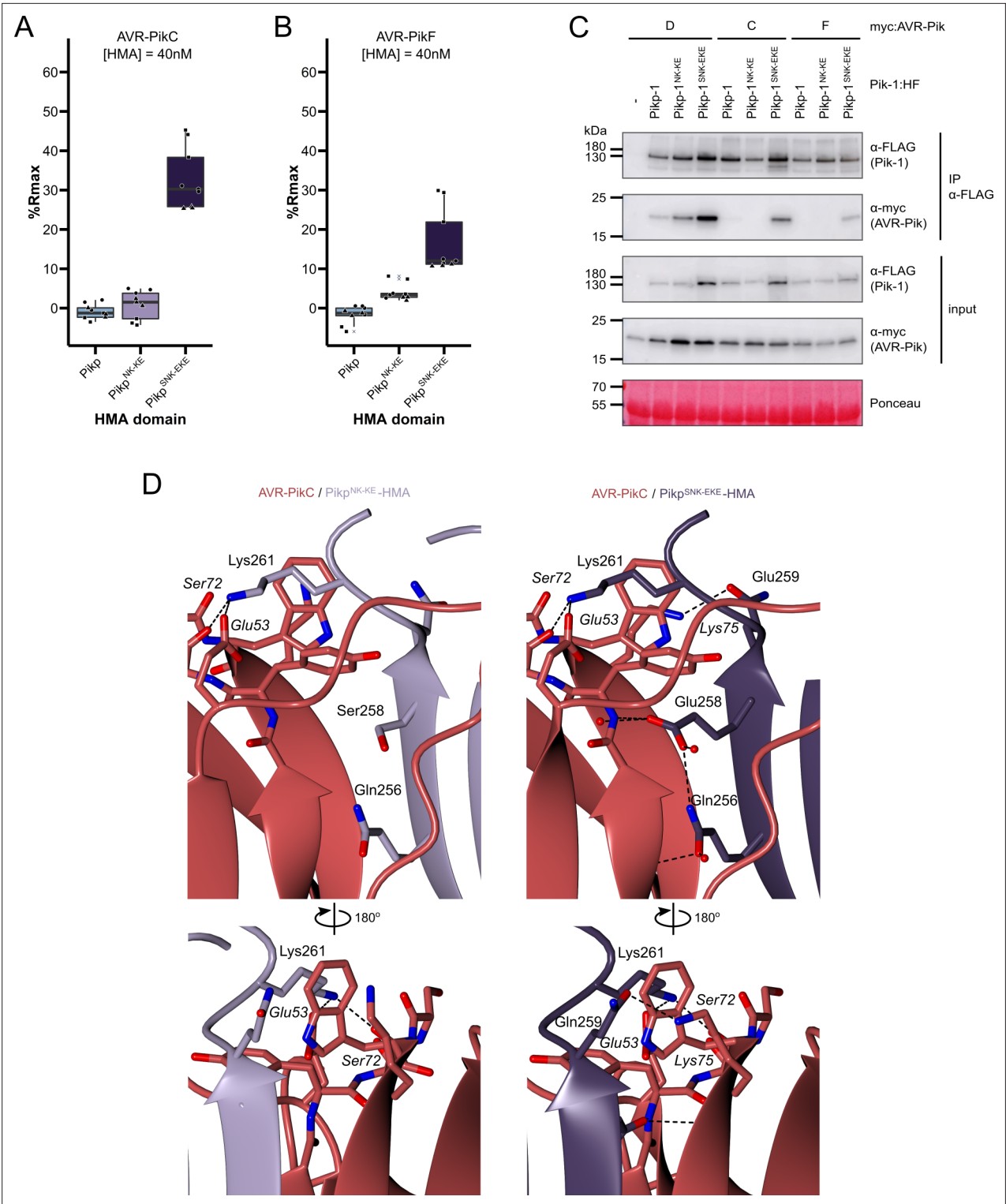

**Figure 3.** The SNK-EKE triple mutation extends Pikp-1 binding to AVR-PikC and AVR-PikF in vitro and in planta by facilitating new contacts across the protein:protein interface. Boxplots showing the %Rmax observed for the interactions between AVR-PikC (**A**) or AVR-PikF (**B**), both at 40 nM injection concentration, and each of Pikp-HMA, Pikp-HMA$^{NK-KE}$, and Pikp-HMA$^{SNK-EKE}$. %Rmax is the percentage of the theoretical maximum response, assuming a 2:1 binding model (as previously observed for Pikp-HMA proteins). The centre line of the box represents the median and the box limits are the upper and lower quartiles. The whiskers extend to the smallest value within Q1 − 1.5 X the interquartile range (IQR) and the largest value within Q3 +1.5 X IQR. Individual data points are represented as black shapes. The experiment was repeated three times, with each experiment consisting of three technical replicates. Data for 4 nM and 100 nM effector injection concentrations are shown in *Figure 3—figure supplement 1*. (**C**) Western blots following co-

*Figure 3 continued on next page*

*Figure 3 continued*

immunoprecipitation show that the Pikp-1$^{SNK-EKE}$ chimera binds to tested AVR-Pik effector variants in *N. benthamiana*. Plant cell lysates were probed for the expression of Pikp-1/Pikp-1$^{NK-KE}$/Pikp-1$^{SNK-EKE}$ and AVR-Pik effector variants using anti-FLAG and anti-Myc antiserum, respectively. Total protein extracts were visualized by Ponceau Staining. (**D**) The crystal structure of the Pikp-HMA$^{SNK-EKE}$/AVR-PikC complex (PDB entry 7QPX) reveals additional hydrogen bonds at the protein:protein interfaces compared to Pikp-HMA$^{NK-KE}$/AVR-PikC (PDB entry 7A8W). Protein structures are represented as ribbons with relevant side chains displayed as cylinders. Dashed lines indicate hydrogen bonds. Relevant water molecules are represented as red spheres.

The online version of this article includes the following source data and figure supplement(s) for figure 3:

**Source data 1.** Unedited and uncropped blot for *Figure 3C*, IP Pik-1 α-FLAG, Pik-1 α-FLAG, with relevant bands labeled.

**Source data 2.** Unedited and uncropped blot for *Figure 3C*, IP Pik-1 α-FLAG, Pik-1 α-FLAG.

**Source data 3.** Unedited and uncropped blot for *Figure 3C*, IP Pik-1 α-FLAG, AVRPik α-myc, with relevant bands labeled.

**Source data 4.** Unedited and uncropped blot for *Figure 3C*, IP Pik-1 α-FLAG, AVRPik α-myc.

**Source data 5.** Unedited and uncropped blot for *Figure 3C*, input, Pik-1 α-FLAG, with relevant bands labeled.

**Source data 6.** Unedited and uncropped blot for *Figure 3C*, input, Pik-1 α-FLAG.

**Source data 7.** Unedited and uncropped blot for *Figure 3C*, input, AVRPik α-myc, with relevant bands labeled.

**Source data 8.** Unedited and uncropped blot for *Figure 3C*, input, AVRPik α-myc.

**Source data 9.** Unedited and uncropped blot for *Figure 3C*, Ponceau stain, with relevant bands labeled.

**Source data 10.** Unedited and uncropped blot for *Figure 3C*, Ponceau stain.

**Source data 11.** Surface plasmon resonance (SPR) data used for box plots in panel A.

**Source data 12.** Surface plasmon resonance (SPR) data used for box plots in panel B.

**Figure supplement 1.** Boxplots showing the %Rmax observed for the interactions between AVR-PikC (**A**) or AVR-PikF (**B**), both at 4 nM and 100 nM injection concentrations, and each of Pikp-HMA, Pikp-HMA$^{NK-KE}$, and Pikp-HMA$^{SNK-EKE}$.

**Figure 3-Figure supplement 1-Source data 1.** Surface plasmon resonance (SPR) data used for box plots in panel A (4 nM).

**Figure 3-Figure supplement 1-Source data 2.** Surface plasmon resonance (SPR) data used for box plots in panel A (100 nM).

**Figure 3-Figure supplement 1-Source data 3.** Surface plasmon resonance (SPR) data used for box plots in panel B (4 nM).

**Figure 3-Figure supplement 1-Source data 4.** Surface plasmon resonance (SPR) data used for box plots in panel B (100 nM).

**Figure supplement 2.** Schematic representation of the crystal structure of the complex formed between Pikp-HMA$^{NK-KE}$ and AVR-PikC (PDB entry 7A8W).

**Figure supplement 3.** Schematic representation of the crystal structure of the complex formed between Pikp-HMA$^{SNK-EKE}$ and AVR-PikC (PDB entry 7QPX).

**Figure supplement 4.** Schematic representation of the crystal structure of the complex formed between Pikp-HMA$^{SNK-EKE}$ and AVR-PikF (PDB entry 7QZD).

Pikp-1$^{SNK-EKE}$ were each co-expressed with either AVR-PikD, AVR-PikC, or AVR-PikF in *N. benthamiana*. As before, Pikp-2 was not included in the co-immunoprecipitation assays to prevent the onset of cell death. We found that AVR-PikD, AVR-PikC, or AVR-PikF co-immunoprecipitated with Pikp-1$^{SNK-EKE}$ (*Figure 3c*); while the band corresponding to AVR-PikF was faint, this can be attributed to lower levels of AVR-PikF in the input. As previously observed (*De la Concepcion et al., 2019*), AVR-PikD, but not AVR-PikC or AVR-PikF co-immunoprecipitated with Pikp-1 and Pikp-1$^{NK-KE}$ (*Figure 3c*). Taken together, the results from in vitro and in planta assays indicate that the Ser258Glu mutation increases the binding of Pikp-1$^{NK-KE}$ for AVR- PikC and AVR-PikF to a sufficient level to trigger cell death in planta.

## Crystal structures of the Pikp-HMA$^{SNK-EKE}$/AVR-PikC and Pikp-HMA$^{SNK-EKE}$/AVR-PikF complexes reveal new contacts across the binding interface

To confirm that the side chain of Glu258 in Pikp-1$^{SNK-EKE}$ forms a new hydrogen bond across the interface (as observed for Glu72 in the OsHIPP19/AVR-PikF complex), we determined the crystal structures of the Pikp-HMA$^{SNK-EKE}$/AVR-PikC and Pikp-HMA$^{SNK-EKE}$/AVR-PikF complexes. For comparison, we also determined the crystal structure of Pikp-HMA$^{NK-KE}$/AVR-PikC. These were produced by co-expression in *E. coli* and purified to homogeneity using established methods for purification of HMA domain/AVR-Pik complexes (*Maidment et al., 2021*; *Maqbool et al., 2015*; *De la Concepcion et al., 2018*; *De la Concepcion et al., 2019*). Crystals were obtained in several conditions in the Morpheus screen (Molecular Dimensions), and X-ray diffraction data were collected at the Diamond Light Source

(Oxford, UK) to a resolution of 2.15 Å (Pikp-HMA$^{NK-KE}$/AVR-PikC), 2.05 Å (Pikp-HMA$^{SNK-EKE}$/AVR-PikC), and 2.2 Å (Pikp-HMA$^{SNK-EKE}$/AVR-PikF). These structures were solved by molecular replacement and refined/validated using standard protocols (see Materials and methods). Data collection, processing, and refinement statistics are shown in *Supplementary file 1*. The final refined models have been deposited at the PDB with accession codes 7A8W, 7QPX, and 7QZD.

The global structures of the complexes are essentially identical to each other and to the previously determined Pik-HMA/AVR-Pik crystal structures (*Figure 3—figure supplement 2*, *Figure 3—figure supplement 3*, *Figure 3—figure supplement 4*, *Supplementary file 1*). The RMSDs, as calculated in COOT with secondary structure matching, between Pikp-HMA$^{NK-KE}$/AVR-PikC and Pikp-HMA$^{SNK-EKE}$/AVR-PikC or Pikp-HMA$^{SNK-EKE}$/AVR-PikF are 0.38 Å using 154 residues and 0.60 Å using 155 residues, respectively. Interface analysis performed with qtPISA (*Krissinel and Henrick, 2009*) identified 15 hydrogen bonds and nine salt bridges between Pikp-HMA$^{SNK-EKE}$ and AVR-PikC, and 16 hydrogen bonds and 11 salt bridges between Pikp-HMA$^{SNK-EKE}$ and AVR-PikF, compared to the 12 hydrogen bonds and eight salt bridges mediating the interaction between Pikp-HMA$^{NK-KE}$ and AVR-PikC (*Supplementary file 2*). Inspection of the structures revealed that the side chain of Glu258 does indeed extend across the interface, forming direct hydrogen bonds with the backbone of the effectors (*Figure 3d*). This single mutation also supports additional hydrogen bonds at the interface between AVR-PikC (or AVR-PikF) and residues comprising β4 of the HMA domain (*Figure 3d*). These differences at interface 3 likely explain the increased binding affinity of Pikp-HMA$^{SNK-EKE}$ for AVR-PikC/AVR-PikF relative to Pikp-HMA$^{NK-KE}$.

## Rice plants expressing Pikp$^{OsHIPP19-mbl7}$ or Pikp-1$^{SNK-EKE}$ are resistant to *M. oryzae* expressing AVR-PikC or AVR-PikF

To determine whether the engineered Pik NLRs Pikp-1$^{OsHIPP19-mbl7}$ and Pikp-1$^{SNK-EKE}$ could mediate resistance to *M. oryzae* Sasa2 isolates expressing either AVR-PikC or AVR-PikF, we generated transgenic rice (*Oryza sativa* cv. Nipponbare, *pikp-*) expressing either wild-type *Pikp-1*, *Pikp-1$^{OsHIPP19-mbl7}$* or *Pikp-1$^{SNK-EKE}$*, with *Pikp-2*. Expression of both *Pikp-1* (or engineered *Pikp-1* variants) and *Pikp-2* were under the control of the constitutive CaMV 35 S promoter and confirmed by RT-PCR (*Figure 4—figure supplement 1*). Rice leaf blade punch inoculation assays were performed to determine resistance to *M. oryzae* (Sasa2) transformants carrying *AVR-Pik* alleles (*AVR-PikC, -PikD* or *-PikF*) or *AVR-Pii* (negative control, AVR-Pii is not recognized by Pikp) in T$_1$ progenies derived from one *Pikp-1/Pikp-2*, six *Pikp-1$^{OsHIPP19-mbl7}$/Pikp-2* and five *Pikp-1$^{SNK-EKE}$/Pikp-2* independent transgenic T$_0$ lines (*Figure 4*, *Figure 4—figure supplement 2*, *Figure 4—figure supplement 3*, *Figure 4—figure supplement 4*). The T$_1$ transgenic rice lines expressing *Pikp-1/Pikp-2* showed resistance to Sasa2 transformed with AVR-PikD, but not to the other transformants (*Figure 4*, *Figure 4—figure supplement 2*). In contrast, the T$_1$ transgenic rice lines expressing either Pikp-1$^{OsHIPP19-mbl7}$/Pikp-2 or Pikp-1$^{SNK-EKE}$/Pikp-2 were resistant to Sasa2 transformed with either AVR-PikC or AVR-PikF, as well as to Sasa2 transformed with AVR-PikD (*Figure 4*, *Figure 4—figure supplement 3*, *Figure 4—figure supplement 4*).

## Discussion

Plant diseases cause significant crop losses and constrain global food production. To develop disease-resistant crops, breeding programs exploit resistance genes present in wild germplasm that can be introgressed into elite cultivars. While recent advances have accelerated efforts to identify and clone resistance genes (*Mascher et al., 2014*; *Arora et al., 2019*; *Jupe et al., 2013*; *Gardiner et al., 2020*), conventional breeding approaches are constrained by the recognition profiles of resistance genes present in wild germplasm. Rational engineering of NLR immune receptors has the potential to yield novel disease resistance traits and expand the repertoire of resistance genes available to combat plant pathogens. It also offers the potential to restore disease resistance that has been overcome by pathogens and accelerate responses to dynamic changes in pathogen-effector populations. Here, we took two approaches to engineer the integrated HMA domain of the NLR protein Pik-1 to deliver new-to-nature effector recognition profiles.

The stealthy effector variants AVR-PikC and AVR-PikF do not interact with the integrated HMA domains of any Pik alleles characterized to date with sufficiently high affinity to activate defense. By contrast, as a putative virulence target of AVR-Pik, OsHIPP19 is bound by all effector variants,

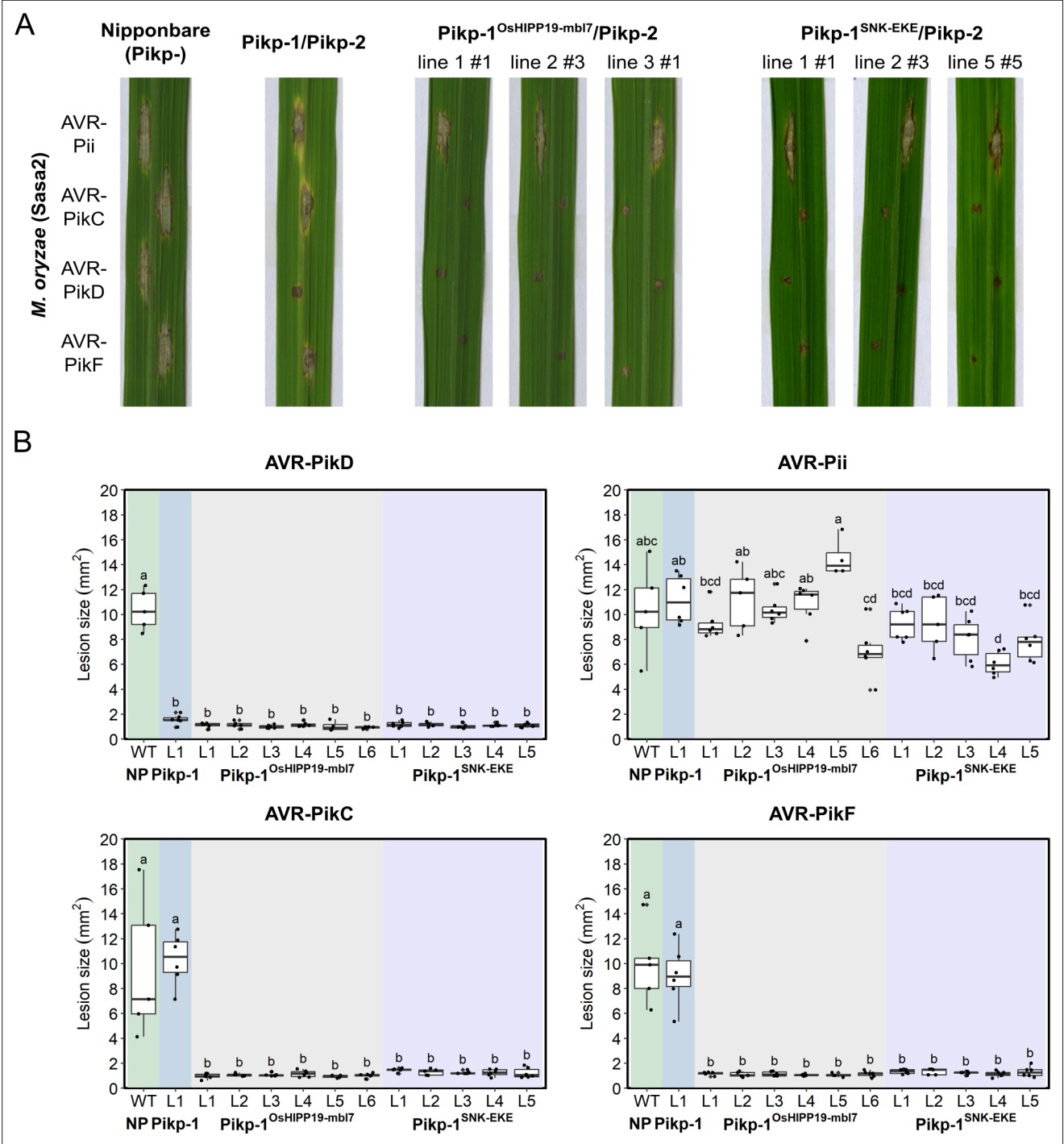

**Figure 4.** Transgenic rice plants carrying the *Pikp-1^OsHIPP19mbl7* chimera or the *Pikp-1^SNK-EKE* mutation show extended resistance to *Magnaporthe oryzae* carrying *AVR-PikC* or *AVR-PikF* compared to Pikp-1 wild-type. (**A**) Example leaves from pathogenicity assays of wild-type *O. sativa* cv. Nipponbare and three transgenic lines of *O. sativa* cv. Nipponbare expressing *Pikp-1/Pikp-2*, *Pikp-1^OsHIPP19mbl7/Pikp-2*, or *Pikp-1^SNK-EKE/Pikp-2* challenged with *M. oryzae* Sasa2 transformed with *AVR-Pii*, *AVR-PikC*, *AVR-PikD*, or *AVR-PikF*. The $T_1$ generation seedlings were used for the inoculation test. Wild-type *O. sativa* cv. Nipponbare (recipient) is susceptible to all *M. oryzae* Sasa2 transformants (left), while the *Pikp-1/Pikp-2* transformant is only resistance to *M. oryzae* Sasa2 transformed with *AVR-PikD* (no development of disease lesions). The *Pikp-1^OsHIPP19mbl7/Pikp-2* or *Pikp-1^SNK-EKE/Pikp-2* plants show resistance to *M. oryzae* Sasa2 transformed with *AVR-PikC, AVR-PikD* or *AVR-PikF* but not *AVR-Pii*. (**B**) Disease lesion sizes (determined using ImageJ) are represented as box plots. L1-L6 and L1-L5 represent data from rice lines as shown in **Figure 4—figure supplements 2–4**. The centre line of the box represents the median and the box limits are the upper and lower quartiles. The whiskers extend to the smallest value within Q1 − 1.5 X the interquartile range (IQR) and the largest value within Q3 + 1.5 X IQR. Individual measurements are represented as black dots. Leaf images corresponding to each of the data

*Figure 4 continued on next page*

*Figure 4 continued*

points presented in the box plots are shown in *Figure 4—figure supplements 2–4*. Lowercase letters represent statistically significant differences (p<0.05, ANOVA with post-hoc Tukey HSD test). RT-PCR confirmed expression of transgenes in (**A**) is shown in *Figure 4—figure supplement 1*.

The online version of this article includes the following source data and figure supplement(s) for figure 4:

**Source data 1.** Lesion size data for pathogen growth on rice plants.

**Figure supplement 1.** RT-PCR to confirm transgene expression of *Pikp-1* (or engineered *Pikp-1* variants) and *Pikp-2* for the individuals shown in *Figure 4*.

**Figure supplement 2.** Pathogenicity assays in the $T_1$ progenies derived from a *Pikp-1/Pikp-2* $T_0$ transgenic line of *O. sativa* cv. Nipponbare against *M. oryzae* Sasa2 transformed with *AVR-Pii*, *AVR-PikC*, AVR-*PikD*, or AVR-*PikF*.

**Figure supplement 3.** Pathogenicity assays in the $T_1$ progenies derived from six independent *Pikp-1$^{OsHIPP19-mbl7}$/Pikp-2* $T_0$ transgenic lines of *O. sativa* cv. Nipponbare against *M. oryzae* Sasa2 transformed with *AVR-Pii*, *AVR-PikC*, AVR-*PikD*, or AVR-*PikF*.

**Figure supplement 4.** Pathogenicity assays in the $T_1$ progenies derived from five independent *Pikp-1$^{SNK-EKE}$/Pikp-2* $T_0$ transgenic lines of *O. sativa* cv. Nipponbare against *M. oryzae* Sasa2 transformed with *AVR-Pii*, *AVR-PikC*, AVR-*PikD*, or AVR-*PikF*.

including AVR-PikC and AVR-PikF with high affinity (*Maidment et al., 2021*; *Oikawa et al., 2020*). Using this knowledge, and the relationship between OsHIPP19 and integrated Pik-HMA domains, we engineered two Pik-1 variants, Pikp-1$^{OsHIPP19-mbl7}$, and Pikp-1$^{SNK-EKE}$. These engineered Pik-1 proteins bound AVR-PikC and AVR-PikF, activated cell death in *N. benthamiana*, and conferred blast resistance in rice. Engineering an NLR-integrated domain to resemble an effector target reduces the likelihood of the effector mutating to evade immune detection while retaining host target binding. Therefore, this approach may represent a route to more durable disease resistance, particularly in the case of effectors whose function is essential for pathogen virulence.

Despite the structural similarity of the OsHIPP19 and Pikp-1 HMA domains, the Pikp-1$^{OsHIPP19}$ chimera triggered effector-independent cell death in *N. benthamiana* when expressed with Pikp-2. This autoactivity required intact P-loop and MHD motifs in Pikp-2, as previously observed for the effector-dependent response of Pikp-1/Pikp-2 to AVR-PikD (*Zdrzałek et al., 2020*). Interestingly, mutating the conserved lysine in the P-loop of Pikp-1$^{OsHIPP19}$ to arginine partially attenuated the cell death response. Mutating the P-loop of Pikp-1 has previously been shown to abolish the effector-dependent cell death response; however, maintenance of effector-independent cell death, albeit at a reduced level, by Pikp-1$^{OsHIPP19\_K296R}$, suggests that an intact P-loop in Pikp-1 is not essential for Pik-mediated signaling. Based on a previously published approach to remove autoactivation when incorporating different HMA domains into Pik-1 (*Białas et al., 2021*), we reverted seven amino acids in the β1-α1 loop of Pikp-1$^{OsHIPP19}$ to those found in Pikp-1, giving the modified chimera Pikp-1$^{OsHIPP19-mbl7}$. The β1-α1 loop contains the classical MxCxxC metal-binding motif which is characteristic of HMA domains and is degenerate in both Pikp-1 (MEGNNC) and OsHIPP19 (MPCEKS). We speculate that this loop is involved in intra- or inter-molecular interactions of Pik-1/Pik-2 which support an inactive state in the absence of effector binding, which is disturbed in the Pikp-1$^{OsHIPP19}$ chimera but restored in Pikp-1$^{OsHIPP19-mbl7}$. These results highlight the potential challenges of incorporating domains that have not co-evolved with other domains in the receptor, but also show the potential for overcoming auto-activation to deliver functional NLRs. A recent study demonstrated that the Pik-1 chassis can accommodate nanobodies to GFP and mCherry, and these mediated reduced viral loads of Potato Virus X (PVX) expressing these antigens in *N. benthamiana* (*Kourelis et al., 2023*). This further demonstrates the potential of the Pik-1 NLR as a versatile system for engineering disease resistance through domain exchange.

Based on the OsHIPP19/AVR-PikF complex (*Maidment et al., 2021*), we incorporated a Ser258Glu point mutation in the Pikp-1$^{NK-EKE}$ background, generating a Pikp-HMA triple mutant, Pikp-1$^{SNK-EKE}$. By determining the crystal structures of the Pikp-HMA$^{SNK-EKE}$/AVR-PikC and Pikp-HMA$^{SNK-EKE}$/AVR-PikF complexes, we confirmed the formation of new contacts across the HMA/effector interface that likely account for the expanded recognition profile to AVR-PikC and AVR-PikF. In addition to a new hydrogen bond between the side chain of Pikp-1$^{SNK-EKE\_Glu258}$ and the backbone of AVR-PikC, we observed two additional intermolecular hydrogen bonds formed between other amino acids at the interface. This extended hydrogen bonding is facilitated by a shift in β4 of the HMA domain towards the effector. Together with previous studies in the Pik-1/Pik-2 and RGA5/RGA4 systems (*De la Concepcion et al., 2019*; *Cesari et al., 2022*; *Liu et al., 2021*), our new results show the utility of

structure-guided approaches to engineering NLR integrated domains to extend binding to different effectors. While recent advances in protein structure modeling will support future engineering efforts, challenges remain in the accurate prediction of side-chain positions, and the effect of individual mutations, which will necessitate experimental determination of protein complexes to optimize intermolecular interactions.

The Pikp-1[SNK-EKE] variant differs from the wild-type Pikh-1 allele in just two amino acid positions. Generating Pikp-1[SNK-EKE] from Pikh-1 requires a maximum of four nucleotide substitutions, which can be achieved using precise base editing and prime editing technologies (*Molla et al., 2021*; *Hua et al., 2022*). In many countries, edited crop varieties which do not contain DNA from another species are not subject to restrictions beyond those required for conventionally bred crop varieties. Therefore, this work raises the exciting prospect of editing wild-type alleles of NLRs that have greater potential for deployment in the field than those incorporating entirely new protein domains or substantial sequence changes.

Given the limited number of *M. oryzae* effectors for which host targets have been identified, it is notable that three (AVR-Pik, AVR-Pia and AVR1-CO39) interact with HMA domains of HIPPs and/or HPPs (*Maidment et al., 2021*; *Oikawa et al., 2020*). The AVR-Pik-like effector APikL2, which is highly conserved across *M. oryzae* lineages with different grass hosts, also binds to the HMA domain of a HIPP (sHMA94 from *Setaria italica*) (*Bentham et al., 2021*). Although interaction with AVR-Pik appears to stabilize HIPPs (*Oikawa et al., 2020*), at present, the consequences and significance for pathogen virulence of these effector/HMA interactions are unclear. Intriguingly, the potato mop-top virus movement protein has been shown to interact with NbHIPP26 (*Cowan et al., 2018*), and other HIPPs/HPPs have been described as host susceptibility factors (*Fukuoka et al., 2009*; *Radakovic et al., 2018*; *Zschiesche et al., 2015*) and may represent targets of effectors from other pathogens. This raises the possibility that incorporating different HMA domains into the Pik-1 chassis (with the mutations described here to prevent autoactivation, if necessary), could offer a suite of NLR proteins capable of recognizing as-yet unknown effectors from diverse pathogens. Alongside biochemical approaches to identify effector-target interactions, advances in structural modeling could also enable the identification of novel HMA-binding effectors. AVR-Pik, AVR-Pia, and AVR1-CO39 all share the conserved MAX structural fold (*de Guillen et al., 2015*), and effectors with a similar structural core may bind HMAs. For example, an integrated HMA domain has been engineered to respond to the MAX effector AVR-Pib (*Liu et al., 2021*), although it is yet to be demonstrated that this effector binds host HMA targets. Identification of specific effector/HMA pairs could guide Pik-1 engineering for the recognition of new pathogens, and potentially enable the design of synthetic HMA domains capable of binding to and recognizing a broad range of pathogen effectors.

Advances in our understanding of the molecular and structural basis of NLR activation have progressed efforts for the rational engineering of NLR proteins with altered recognition profiles. Previous studies have successfully engineered integrated domains to extend recognition capacities, though so far this has either resulted in the regeneration of resistance already conferred by other NLRs (*De la Concepcion et al., 2019*; *Cesari et al., 2022*; *Liu et al., 2021*) or provided recognition of a protein not present in the native pathogen (*Kourelis et al., 2023*). In this study, we use an effector target to guide the engineering of an integrated domain to deliver two engineered Pik-1 variants with new-to-nature effector recognition profiles. The chimeric NLR Pikp-1[OsHIPP19-mbl7] highlights the potential to incorporate diverse HMA domains without rendering the chimera autoactive. The triple mutant Pikp-1[SNK-EKE] illustrates the benefit of structural/biochemical characterization of effector-target interactions to inform rational engineering. Crucially, both engineered NLR proteins deliver novel resistance in transgenic rice, and have potential for deployment in the field against *M. oryzae* isolates carrying the stealthy AVR-PikC and AVR-PikF alleles. This study demonstrates the value of target-guided approaches in engineering NLR proteins with new-to-nature recognition profiles. We propose that this approach could expand the 'toolbox' of resistance genes to counter the devastating impacts of plant pathogens on crop yields and global food security.

## Materials and methods

### Gene cloning for protein expression in *N. benthamiana*

For protein expression in planta, full-length Pikp-1 NLRs containing the OsHIPP19 HMA domain (and Pikp-1[OsHIPP43-mbl7]), the Pikp-1[SNK-EKE] mutation (made by introducing the S258E mutation in the HMA domain by PCR), and other HMA domain mutations were assembled using Golden Gate cloning into the plasmid pICH47742 with a C-terminal 6xHis/3xFLAG tag. Expression was driven by the *A. tumefaciens* Mas promoter and terminator. Full-length wild-type Pikp-1, Pikp-1[NK-KE], Pikp-2, and AVR-Pik variants used were generated as described previously (*Maqbool et al., 2015*; *De la Concepcion et al., 2018*; *De la Concepcion et al., 2021*; *Longya et al., 2019*; *De la Concepcion et al., 2019*). All DNA constructs were verified by sequencing.

### Gene cloning, expression, and purification of proteins for in vitro binding studies

For SPR, Pikp-HMA[NK-KE] and Pikp-HMA[SNK-EKE] (residues Gly186 – Asp264) variants were cloned into pOPIN-M (generating a 3 C protease cleavable N-terminal 6xHis:MBP-tag). AVR-PikD, AVR-PikC, and AVR-PikF (residues Glu22 – Phe93) were cloned into pOPIN-E (generating a C-terminal non-cleavable 6xHis-tag, but also including a 3 C protease cleavable N-terminal SUMO-tag, as detailed previously *Maqbool et al., 2015*). The Pikp-HMA[NK-KE] and Pikp-HMA[SNK-EKE] proteins were expressed and purified using the same pipeline as described below for obtaining protein complexes for crystallization, whereas the effectors were retained on the second pass through the 5 ml Ni²⁺-NTA column (which served to remove the SUMO tag following 3 C cleavage) requiring specific elution with elution buffer (50 mM Tris-HCl pH 8.0, 50 mM glycine, 0.5 M NaCl, 500 mM imidazole, 5% (v/v) glycerol), followed by gel filtration using a Superdex 75 26/600 column equilibrated in running buffer (20 mM HEPES pH 7.5 and 150 mM NaCl). Proteins were concentrated and stored at –80°C for further studies.

### Cloning, expression, and purification of proteins for crystallization

For crystallization of the Pikp-HMA[NK/KE]/AVR-PikC, Pikp-HMA[SNK/EKE]/AVR-PikC and Pikp-HMA[SNK/EKE]/AVR-PikF complexes, Pikp-HMA (residues Gly186 – Asp264) variants were cloned into pOPIN-M and AVR-PikC or AVR-PikF into pOPIN-A using InFusion cloning. Chemically competent *E. coli* SHuffle cells (*Lobstein et al., 2012*) were co-transformed with these vectors to produce 6xHis-MBP-tagged Pikp-HMA domains and untagged effectors. Cultures of these cells were grown in auto-induction media (*Studier, 2005*) to an $OD_{600}$ of 0.4–0.6 at 30 °C, then incubated overnight at 18 °C. Cells were harvested by centrifugation and resuspended in lysis buffer (50 mM Tris-HCl pH 8.0, 50 mM glycine, 0.5 M NaCl, 20 mM imidazole, 5% (v/v) glycerol, 1 cOmplete EDTA-free protease inhibitor cocktail tablet (Roche) per 50 ml buffer). Resuspended cells were lysed by sonication with a VibraCell sonicator (SONICS), and the whole-cell lysate was clarified by centrifugation. An AKTA Xpress (GE Healthcare) system was used to carry out a two-step purification at 4 °C. The clarified cell lysate was first injected into a 5 ml Ni²⁺-NTA column (GE Healthcare). The complexes were step-eluted with elution buffer (50 mM Tris-HCl pH 8.0, 50 mM glycine, 0.5 M NaCl, 500 mM imidazole, 5% (v/v) glycerol), and directly applied to a Superdex 75 26/600 gel filtration column equilibrated in a running buffer (20 mM HEPES pH 7.5 and 150 mM NaCl). The 6xHis-MBP tag was cleaved from Pik-HMA by incubation with 3 C protease (1 µg protease per mg of fusion protein) at 4 °C overnight and removed by passing the sample through a 5 ml Ni²⁺-NTA column connected to a 5 ml dextrin sepharose (MBPTrap) column (GE Healthcare), both equilibrated in lysis buffer. Fractions containing the relevant complexes were then concentrated and injected onto a Superdex 75 26/600 column equilibrated in running buffer. Eluted fractions containing protein complexes were concentrated to 13 mg/mL (Pikp-HMA[NK/KE]/AVR-PikC), 10 or 20 mg/mL (Pikp-HMA[SNK/EKE]/AVR-PikC) and 20 mg/mL (Pikp-HMA[SNK/EKE]/AVR-PikF) for crystallization. Protein concentrations were determined using a Direct Detect Infrared Spectrometer (Millipore Sigma).

### Protein crystallization, data collection, structure solution, refinement, and validation

Crystallization trials were set up in 96-well plates using an Oryx Nano robot (Douglas Instruments) with 0.3 µl of protein combined with 0.3 µl reservoir solution. Crystals of each complex were obtained

in multiple conditions using the commercially available Morpheus screen (Molecular Dimensions). Crystals used for X-ray data collection were obtained from condition F8 (Pikp-HMA$^{NK/KE}$/AVR-PikC complex), D7 (Pikp-HMA$^{SNK/EKE}$/AVR-PikC), and H4 (Pikp-HMA$^{SNK/EKE}$/AVR-PikF). The crystals were snap-frozen in liquid nitrogen and shipped to Diamond Light Source for X-ray data collection. Diffraction data were collected at the Diamond Light Source, i04, and i03 beamlines (see *Supplementary file 1*), under proposals mx13467 and mx18565. The data were scaled and merged by Aimless in the CCP4i2 software package (*Potterton et al., 2018*). Each of the structures was solved by molecular replacement using PHASER (*McCoy et al., 2007*). The search models used were the Pikp-HMA/AVR-PikD complex (PDB entry: 5A6W) for Pikp-HMA$^{NK/KE}$/AVR-PikC, the Pikp-HMA$^{NK/KE}$/AVR-PikC complex (PDB entry: 7A8W) for Pikp-HMA$^{SNK/EKE}$/AVR-PikC, and the Pikp-HMA/AVR-PikD complex (PDB entry: 5A6W) for Pikp-HMA$^{SNK/EKE}$/AVR-PikF. Iterative cycles of the manual model building using COOT (*Emsley and Cowtan, 2004*) and refinement with REFMAC (*Murshudov et al., 2011*) were used to derive the final structures, which were validated using the tools in COOT and MolProbity (*Williams et al., 2018*). The final protein structures, and the data used to derive them, have been deposited at the Protein Data Bank with IDs 7A8W (Pikp-HMA$^{NK/KE}$/AVR-PikC), 7QPX (Pikp-HMA$^{SNK/EKE}$/AVR-PikC), and 7QZD (Pikp-HMA$^{SNK/EKE}$/AVR-PikF).

## In vitro protein-protein interaction studies: SPR

SPR was performed using a Biacore T200 (Cytiva) at 25 °C and at a flow rate of 30 μl/min. The running buffer was 20 mM HEPES pH 7.5, 860 mM NaCl, and 0.1%(v/v) Tween20. Flow cell (FC) 2 of an NTA chip (GE Healthcare) was activated with 30 μl 0.5 mM NiCl$_2$. 30 μl of the 6xHis-tagged effector (the ligand) was immobilized on FC2 to give a response of ~250 RU. The HMA domain (the analyte) was then flowed over both FC1 and FC2 for 360 s, followed by a dissociation time of 180 s. Three separate concentrations of each HMA were tested, 4 nM, 40 nM, and 100 nM. The NTA chip was regenerated after each cycle with 30 μl 0.35 M EDTA pH 8.0. The background response from FC1 (non-specific binding of the HMA domain to the chip) was subtracted from the response from FC2. To obtain %R$_{max}$, the binding response (R$_{obs}$) was measured immediately prior to the end of injection and expressed as a percentage of the theoretical maximum response (R$_{max}$) assuming a 2:1 HMA:effector binding model for Pikp-HMA, Pikp$^{NK-KE}$-HMA, and Pikp$^{SNK-EKE}$-HMA calculated as follows:

$$R_{max}\left(RU\right) = \frac{M_W(analyte)}{M_W(ligand)} \times stoichiometry \times ligandcapture(RU)$$

Data analysis and visualization was carried out in R v4.1.2 (*R Core Development Team, 2018*) using the packages dplyr (v1.0.9 [*Wickham, 2022*]) and ggplot2 (v3.3.6 [*Wickham, 2016*]).

## *N. benthamiana* cell death assays

*A. tumefaciens* GV3101 (C58 (rifR) Ti pMP90 (pTiC58DT-DNA) (gentR) Nopaline(pSouptetR)) cells carrying relevant Pikp-1 constructs, were resuspended in agroinfiltration media (10 mM MES pH 5.6, 10 mM MgCl2 and 150 μM acetosyringone) and mixed with *A. tumefaciens* GV3101 carrying Pikp-2, AVR-Pik effectors, and P19 at OD$_{600}$ 0.4, 0.4, 0.6, and 0.1, respectively. 4-week-old *N. benthamiana* leaves were infiltrated using a needleless syringe. *N. benthamiana* plants were grown in a controlled environment room at 22 °C constant temperature and 80% relative humidity, with a 16 hr photoperiod, and were returned to the same room following infiltration. Leaves were collected at 5 dpi (days post infiltration) and photographed under visible and UV light. Images shown are representative of three independent experiments, with a minimum of ten repeats (leaves) in each experiment. The cell death index used for scoring is as presented previously (*De la Concepcion et al., 2018*). Data analysis and visualization were carried out in R v4.1.2 (*R Core Development Team, 2018*) using the packages dplyr (v1.0.9 [*Wickham, 2022*]) and ggplot2 (v3.3.6 [*Wickham, 2016*]). Relevant comparisons between conditions (i.e. combinations of NLRs and effectors) in cell death assays were made using estimation methods (*Ho et al., 2019*) using the package besthr (v0.2.0 [*MacLean, 2022*]). Mean scores were calculated for each condition and each biological replicate. These means were ranked, and a mean rank was calculated for each condition. Bootstrapping was then used to estimate the confidence interval of the mean rank of each condition. 1000 samples were drawn (with replacement) from the values present in the dataset, and the mean rank was calculated for each of these samples to give a distribution of mean rank estimates. The 2.5 and 97.5 quantiles were determined. Conditions are considered to be different if their means lie outside the 2.5 and 97.5 quantiles.

## Confirmation of protein production in cell death assays by western blot analysis

Western blot analysis was used to confirm the presence of proteins in *N. benthamiana* during cell death assays. Three leaf discs were taken at 2 dpi, flash-frozen in liquid nitrogen, and ground to a fine powder using a micropestle. Leaf powder was mixed with 300 µl of plant protein extraction buffer (GTEN (25 mM Tris-HCl, pH 7.5, 150 mM NaCl, 1 mM EDTA, 10 % v/v glycerol), 10 mM DTT, 2% (w/v) PVPP, 0.1% Tween–20, 1 x plant protease inhibitor cocktail (Sigma)). The sample was clarified by centrifugation (20,000 × *g* for 2 min at 4 °C, twice) and 40 µL of the resulting supernatant was added to 10 µL SDS-PAGE loading dye (RunBlue 4 x LDS sample buffer (Expedeon)), followed by incubation at 95 °C for 5 min.

Samples were subjected to SDS-PAGE/western blot analysis to detect epitope-tagged proteins. Pikp-1, Pikp-2, and AVR-Pik effectors were detected by probing membranes with anti-FLAG-HRP (Generon, 1:10,000 dilution), anti-HA-HRP (Thermo Fisher Scientific, 1:3000 dilution) and anti-Myc-HRP (Santa Cruz, 1:5000 dilution) antibodies, respectively, and LumiBlue ECL Extreme (Expedeon). Membranes were also stained with Ponceau S to observe protein loading.

## In planta protein-protein interaction studies: co-immunoprecipitation

For co-immunoprecipitation assays, three leaves were harvested at 3 dpi and flash-frozen in liquid nitrogen. Leaf tissue was ground to a fine powder in liquid nitrogen using a pre-chilled pestle and mortar and resuspended in ice-cold plant protein extraction buffer (2 ml/mg of powder). Plant cell debris was pelleted by centrifugation at 4200 × *g* for 30 min at 4 °C and the supernatant was filtered through a 0.45 µm membrane. 20 µL of filtered extracts were combined with 5 µL SDS-PAGE loading dye as input samples. For immunoprecipitation, 1 mL of filtered protein extract was mixed with 40 µL of anti-FLAG M2 magnetic beads (Merck, formerly Sigma-Aldrich) (equilibrated in GTEN +0.1% Tween-20 prior to use) in a rotary mixer for 1 hr at 4 °C. The beads were washed 5 X in ice-cold IP buffer by separating the beads using a magnetic rack, and the proteins were eluted by resuspending the beads in 30 µL SDS loading buffer and incubating at 70 °C for 10 min. Following SDS-PAGE and transfer, membranes were probed with antibodies as above.

## Fungal strains and transformation

To generate *M. oryzae* Sasa2 harboring different *AVR-Pik* alleles or *AVR-Pii*, Sasa2 was transformed individually with expression vectors for AVR-PikC, -PikD, and -PikF and AVR-Pii. The expression vectors used for generating transgenic *M. oryzae* Sasa2 were pCB1531:*AVR-Pii* promoter:*AVR-Pii* constructed by *Yoshida et al., 2009*, pCB1531:*AVR-PikD* promoter:*AVR-PikC*, -PikD constructed by *Kanzaki et al., 2012*, and pCB1531:*AVR-PikD* promoter:*AVR-PikF*, generated according to *Kanzaki et al., 2012*. The template DNA for AVR-PikF was synthesized by GENEWIZ (Genewiz, Saitama, Japan). These expression vectors were used to transform Sasa2 (lacking *AVR-PikD* alleles and *AVR-Pii*) following the method of *Sweigard et al., 1997*.

## Rice transformation and confirmation of transgene expression

To generate constructs for rice transformation, Golden Gate Level 1 constructs encoding a hygromycin resistance cassette (35 S promoter/nos terminator), untagged *Pikp-1/Pikp-1$^{OsHIPP19-mbl7}$/Pikp-1$^{SNK-EKE}$* with the NLR flanked by the 35 S promoter and terminator, and untagged *Pikp-2* also flanked by the 35 S promoter and terminator were assembled by Golden Gate assembly into the Level 2 vector pICSL4723. The resulting Level 2 constructs contained the hygromycin resistance cassette, 35 S::*Pikp-1/Pikp-1$^{OsHIPP19-mbl7}$/Pikp-1$^{SNK-EKE}$*, and 35 S::*Pikp-2*. These constructs were introduced into *Agrobacterium tumefaciens* (strain EHA105) and used for Agrobacterium-mediated transformation of *Oryza sativa* cv. Nipponbare follows the method of *Okuyama et al., 2011*.

PCR confirmation of the presence of *Pikp-1/Pikp-2* transgenes in T$_1$ progenies used two primer sets, *Pikp-1* (F:TGATCAAAGACCACTTCCGCGTTC +R:TGCTGCCCGCAATGTTTTCACTGC) and *Pikp-2* (F: ATTGTATATGTCAGCCAGAAAATG +R:TCCTCAGGGACTTGCTCGTCTAC) that amplify *Pikp-1* and *Pikp-2*, respectively.

To confirm transgene expression, total RNA was extracted from leaves using an SV Total RNA Isolation System (Promega, WI, USA) and used for RT-PCR. cDNA was synthesized from 500 ng total RNA using a Prime Script RT Reagent Kit (Takara Bio, Otsu, Japan). RT-PCR was performed using three

primer sets, *Pikp-1* (F:TGATCAAAGACCACTTCCGCGTTC +R:TGCTGCCCGCAATGTTTTCACTGC), *Pikp-2* (F:ATTGTATATGTCAGCCAGAAAATG +R:TCCTCAGGGACTTGCTCGTCTAC) and *OsActin* (F: CTGAAGAGCATCCTGTATTG +R: GAACCTTTCTGCTCCGATGG) that amplify *Pikp-1*, *Pikp-2*, and *OsActin*, respectively.

## Disease resistance/virulence assays in rice

Rice leaf blade punch inoculation was performed using the *M. oryzae* isolates. A conidial suspension ($3 \times 10^5$ conidia mL$^{-1}$) was punch inoculated onto a rice leaf one month after seed sowing. The inoculated plants were placed in a dark dew chamber at 27 °C for 24 hr and then transferred to a growth chamber with a 16 hr light/8 hr dark photoperiod. Disease lesions were scanned at 7 days post-inoculation (dpi), and lesion size was measured manually using Image J software (*Schneider et al., 2012*).

## Acknowledgements

This work was supported by the UKRI Biotechnology and Biological Sciences Research Council (BBSRC) Norwich Research Park Biosciences Doctoral Training Partnership, UK [grant BB/M011216/1]; the UKRI BBSRC, UK [grants BB/P012574, BBS/E/J/000PR9795, BB/M02198X], the European Research Council [ERC; proposal 743165]; The Thailand Research Fund through The Royal Golden Jubilee Ph.D. Program [PHD/0152/2556]; the John Innes Foundation; the Gatsby Charitable Foundation; The British Society for Plant Pathology (undergraduate vacation bursary); JSPS KAKENHI 15H05779 and 20H05681; JSPS/The Royal Society Bilateral Research for the project 'Retooling rice immunity for resistance against rice blast disease' (2018–2019). We would also like to thank Julia Mundy and David Lawson from the JIC Biophysical Analysis and X-ray Crystallography platform for their support with protein crystallization and X-ray data collection, Andrew Davies and Phil Robinson from JIC Scientific Photography for their help with leaf imaging, Gerhard Saalbach and Carlo de Oliveira Martins from the JIC Proteomics platform for intact mass spectrometry analysis, and Dan Maclean from The Sainsbury Laboratory for support with statistical analyses. We also thank all members of the Banfield, Kamoun, and Terauchi groups for their discussions.

## Additional information

### Competing interests

Sophien Kamoun: receives funding from industry on NLR biology. The other authors declare that no competing interests exist.

### Funding

| Funder | Grant reference number | Author |
| --- | --- | --- |
| Biotechnology and Biological Sciences Research Council | BB/M011216/1 | Josephine HR Maidment |
| Biotechnology and Biological Sciences Research Council | BB/P012574 | Sophien Kamoun |
| Biotechnology and Biological Sciences Research Council | BBS/E/J/000PR9795 | Sophien Kamoun |
| Biotechnology and Biological Sciences Research Council | BB/M02198X | Marina Franceschetti Mark J Banfield |
| European Research Council | 743165 | Adam R Bentham |
| Royal Golden Jubilee | PHD/0152/2556 | Apinya Longya |
| John Innes Foundation | | Juan Carlos De la Concepcion |

| Funder | Grant reference number | Author |
|---|---|---|
| Gatsby Charitable Foundation | | Aleksandra Białas |
| JIC Excellence with Impact | | Sham Vera |
| Japan Society for the Promotion of Science | KAKENHI 15H05779 and 20H05681 | Motoki Shimizu |
| ERASMUS | | Sham Vera |

The funders had no role in study design, data collection and interpretation, or the decision to submit the work for publication.

## Author contributions

Josephine HR Maidment, Conceptualization, Resources, Formal analysis, Supervision, Validation, Investigation, Visualization, Methodology, Writing – original draft, Project administration, Writing – review and editing; Motoki Shimizu, Conceptualization, Resources, Formal analysis, Validation, Investigation, Visualization, Methodology, Project administration, Writing – review and editing; Adam R Bentham, Resources, Validation, Investigation, Visualization, Methodology, Writing – review and editing; Sham Vera, Juan Carlos De la Concepcion, Resources, Investigation, Methodology, Writing – review and editing; Marina Franceschetti, Supervision, Investigation, Methodology, Writing – review and editing; Apinya Longya, Resources, Investigation, Writing – review and editing; Clare EM Stevenson, Methodology, Writing – review and editing; Aleksandra Białas, Resources, Writing – review and editing; Sophien Kamoun, Ryohei Terauchi, Conceptualization, Supervision, Funding acquisition, Methodology, Project administration, Writing – review and editing; Mark J Banfield, Conceptualization, Supervision, Funding acquisition, Validation, Investigation, Methodology, Writing – original draft, Project administration, Writing – review and editing

## Author ORCIDs

Josephine HR Maidment ![ORCID] http://orcid.org/0000-0002-8229-2718
Motoki Shimizu ![ORCID] http://orcid.org/0000-0002-5622-5554
Adam R Bentham ![ORCID] http://orcid.org/0000-0001-5906-0962
Sham Vera ![ORCID] http://orcid.org/0000-0002-1092-7905
Clare EM Stevenson ![ORCID] http://orcid.org/0000-0001-6695-8201
Juan Carlos De la Concepcion ![ORCID] http://orcid.org/0000-0002-7642-8375
Aleksandra Białas ![ORCID] http://orcid.org/0000-0002-1135-2189
Sophien Kamoun ![ORCID] http://orcid.org/0000-0002-0290-0315
Ryohei Terauchi ![ORCID] http://orcid.org/0000-0002-0095-4651
Mark J Banfield ![ORCID] http://orcid.org/0000-0001-8921-3835

## Decision letter and Author response

Decision letter https://doi.org/10.7554/eLife.81123.sa1
Author response https://doi.org/10.7554/eLife.81123.sa2

# Additional files

### Supplementary files

• Supplementary file 1. Table S1. X-ray data collection and refinement statistics for Pikp-HMA$^{NK-KE}$/AVR-PikC (PDB entry 7A8W), Pikp-HMA$^{SNK-EKE}$/AVR-PikC (PDB entry 7QPX), and Pikp-HMA$^{SNK-EKE}$/AVR-PikF (PDB entry 7QZD).

• Supplementary file 2. Table S2. Summary of interface analysis by QtPISA for Pikp-HMA$^{NK-KE}$/AVR-PikC (PDB entry 7A8W), Pikp-HMA$^{SNK-EKE}$/AVR-PikC (PDB entry 7QPX), and Pikp-HMA$^{SNK-EKE}$/AVR-PikF (PDB entry 7QZD). Protein chains used for the analysis in each complex (as defined in the PDB entries) are: Pikp$^{NK-KE}$:AVR-PikC (E and F); Pikp$^{SNK-EKE}$:AVR-PikC (E and F); Pikp$^{SNK-EKE}$:AVR-PikF (F and G).

• MDAR checklist

### Data availability

All data generated or analysed during this study are included in the manuscript and supporting files.

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
