## [Editor Report]

Engineering NLR proteins to improve disease resistance in crop plants is a major goal of the field. This study applies knowledge from structural and evolutionary studies of the rice NLR protein Pik-1 and cognate effector protein AVR-Pik to engineering of new disease resistance genes. The authors nicely demonstrate that it is indeed possible to engineer resistance proteins with broad recognition specificity for the rice blast fungus. The work is of interest to colleagues in synthetic biology, protein engineering and plant-pathogen interactions.

---

## [Decision Letter]

**Decision letter after peer review:**

Thank you for submitting your article "Effector target-guided engineering of an integrated domain expands the disease resistance profile of a rice NLR immune receptor" for consideration by *eLife*. Your article has been reviewed by 2 peer reviewers, including Jian-Min Zhou as the Reviewing Editor and Reviewer #1, and the evaluation has been overseen by Jürgen Kleine-Vehn as the Senior Editor. The following individual involved in review of your submission has agreed to reveal their identity: Xiaorong Tao (Reviewer #2).

Essential revisions:

1) 35S promoter instead of native promoter was used for transforming rice plants. Ideally, native promoter should be used. The authors need to explain why this was not done. It is essential to provide data on trait analysis of the transgenic lines. Do they grow and develop normally? These are highly important, as Figures1E, 1G, 2E, S3B, S3C, S12B suggest minor HR elicitation by Pikp-1-OsHIPP-mbl7 and Pikp-1-SNK-EKE coexpressed Pikp-2 in the absence of AVR.

2) Quality of some of the co-IP experiments are poor. In Figures1A, 3C, the input of some of the AVR-Pik proteins is too low, and this makes the interpretation of protein-protein interaction data difficult. Also, the anti-FLAG immune blot in Figure 3C is not of sufficient quality. Please provide better blots for these figures.

3) Figures4B, S21, S22, S23, please provide error bars and statistic analyses.

4) Transgenic rice carrying Pikp-1SNK-EKE can respond to AVR-PikD, AVR-PikC, and AVR-PikF. It is necessary to show whether they retain the recognition of AVR-PikA and AVR-PikE. Alternatively this can be done in N. benthamiana plants.

5) The description of Figure 1B "Representative leaf image showing the Pikp-1OsHIPP19 chimera is autoactive in N. benthamiana." This can be misleading, as co-expression of Pikp-1OsHIPP19 with empty vector did not induce cell death. It only did so when co-expressed with Pikp-2.

6) Figure 3D, please label the amino acid residues that form hydrogen bonds.

---

## [Author Response]

Essential revisions:1) 35S promoter instead of native promoter was used for transforming rice plants. Ideally, native promoter should be used. The authors need to explain why this was not done. It is essential to provide data on trait analysis of the transgenic lines. Do they grow and develop normally? These are highly important, as Figures1E, 1G, 2E, S3B, S3C, S12B suggest minor HR elicitation by Pikp-1-OsHIPP-mbl7 and Pikp-1-SNK-EKE coexpressed Pikp-2 in the absence of AVR.

In addition to the constructs detailed here, where the 35S promoter was used, we also transformed rice with Pikp-1 (or one of the engineered variants) and Pikp-2 in a head-to-head orientation under the control of the native promoter. Preliminary results indicated that none of the transformed plants with the native promoter were resistant to *Magnaporthe oryzae* Sasa2 transformed with AVR-PikD, while many plants transformed with constructs using the 35S promoter showed resistance. We therefore continued with the 35S promoter for the experiments presented here with *M. oryzae* transformed with AVR-PikD (positive control), AVR-PikC, AVR-PikF and AVR-Pii (negative control). Future work will investigate whether transgenic plants with the Pik NLRs under the control of the native promoter can also confer resistance. As the reviewer notes, this would be particularly important if a base editing approach was to be taken to engineer Pikp-1^SNK-EKE^.

We agree that future work should carry out a comprehensive assessment of the effect of the transgenes on traits including growth, development, yield, response to other biotic stresses and response to abiotic stress. This information will be crucial if these engineered resistance genes are to be incorporated into a product and deployed in the field. However, we feel that this falls outside the scope of this manuscript, which focuses on the potential of effector target-guided engineering of NLR immune receptors for new disease resistance profiles.

We would not consider the occasional score of 1 or 2 in the cell death experiments in the figures referenced to constitute minor HR elicitation; we observe this in our negative controls, for example Pikp-1^OsHIPP19^ without Pikp-2. Consistent with this, the infection assays carried out with the transgenic rice plants demonstrated that the resistance conferred by the engineered Pik variants is specific to *M. oryzae* transformed with AVR-Pik, as these transgenic plants were susceptible to *M. oryzae* transformed with the unrelated effector AVR-Pii.

2) Quality of some of the co-IP experiments are poor. In Figures1A, 3C, the input of some of the AVR-Pik proteins is too low, and this makes the interpretation of protein-protein interaction data difficult. Also, the anti-FLAG immune blot in Figure 3C is not of sufficient quality. Please provide better blots for these figures.

We have repeated the co-IP experiments in figures 1A, 3C and S9 to obtain better quality images, and have replaced these in the respective figures. We hope the reviewers agree they are improved. When repeating the co-IP experiments with optimised conditions (to improve the quality), we no longer observed interaction of AVR-PikF with the Pikp-1^OsHIPP19^ chimera. This is presented in the new Figure 1—figure supplement 3 (previously Figure 1A, and now also includes AVR-PikF with Pikp-1). In the new Figure 3C, we retain interaction of AVR-PikF with the Pikp-1^SNK-EKE^ construct. With the revised manuscript we also replaced the western blot originally presented as Supplementary Figure 9 to include Pikp-1 with AVR-PikF as a control (and to improve the quality of this blot). We have also exchanged the blots of Pikp-1^OsHIPP19^ and Pikp-1^OsHIPP19-mbl7^ interactions with the effectors as we now think it makes more sense to present the Pikp-1^OsHIPP19-mbl7^ results in the main manuscript. For the narrative this necessitated a re-ordering of Figure 1.

3) Figures4B, S21, S22, S23, please provide error bars and statistic analyses.

We have reviewed the presentation of the data from the transgenic rice experiments to improve clarity and statistical analysis. We have replaced the barplots of original figures 4B, S21, S22 and S23 with the boxplots in figure 4, with each box summarising the data obtained from an independent transgenic event. Statistical analysis was carried out (ANOVA with post-hoc Tukey HSD test) and the results reported in the boxplots shown in figure 4.

4) Transgenic rice carrying Pikp-1SNK-EKE can respond to AVR-PikD, AVR-PikC, and AVR-PikF. It is necessary to show whether they retain the recognition of AVR-PikA and AVR-PikE. Alternatively this can be done in N. benthamiana plants.

The purpose of this paper was to engineer Pik-1 to provide resistance to AVR-Pik variants not recognised by any of the Pik-1 alleles identified in nature to date. For this reason, we did not include AVR-PikA and AVR-PikE in cell death assays and rice transgenic work.

While we maintain that this result is not required to address the main question in this paper, we have now carried out cell death assays with both engineered Pik variants (Pikp-1^SNK-EKE^ and Pikp-1^OsHIPP19_mbl7^) and demonstrated that they both trigger Pikp-2-dependent cell death in response to either AVR-PikA or AVR-PikE. This data is presented in the new figures, Figure 2—figure supplement 2, Figure 2—figure supplement 3, Figure 2—figure supplement 7 and Figure 2—figure supplement 8.

5) The description of Figure 1B "Representative leaf image showing the Pikp-1OsHIPP19 chimera is autoactive in N. benthamiana." This can be misleading, as co-expression of Pikp-1OsHIPP19 with empty vector did not induce cell death. It only did so when co-expressed with Pikp-2.

We have replaced this with “Representative leaf image showing the Pikp-1^OsHIPP19^ chimera is autoactive in *N. benthamiana* in a Pikp-2 dependent manner." This is now shown in Figure 1a.

6) Figure 3D, please label the amino acid residues that form hydrogen bonds.

We have done this.